# Decomposing virulence to understand bacterial clearance in persistent infections

**Beatriz Acuña Hidalgo**[1,4], **Luís M. Silva** [1,2,4], **Mathias Franz** [1,4],
**Roland R. Regoes** [3] **& Sophie A. O. Armitage** [1] ✉

Following an infection, hosts cannot always clear the pathogen, instead either dying or surviving with a persistent infection. Such variation is ecologically and evolutionarily important because it can affect infection prevalence and transmission, and virulence evolution. However, the factors causing variation in infection outcomes, and the relationship between clearance and virulence are not well understood. Here we show that sustained persistent infection and clearance are both possible outcomes across bacterial species showing a range of virulence in *Drosophila melanogaster*. Variation in virulence arises because of differences in the two components of virulence: bacterial infection intensity inside the host (exploitation), and the amount of damage caused per bacterium (per parasite pathogenicity). As early-phase exploitation increased, clearance rates later in the infection decreased, whereas there was no apparent effect of per parasite pathogenicity on clearance rates. Variation in infection outcomes is thereby determined by how virulence – and its components – relate to the rate of pathogen clearance. Taken together we demonstrate that the virulence decomposition framework is broadly applicable and can provide valuable insights into host-pathogen interactions.

Once a host is infected, the immune system may limit pathogen growth, a response termed resistance[1–3]. Reduced pathogen growth can be beneficial to the host by limiting pathogen load and/or facilitating pathogen clearance. Despite such benefits, resistance can have evolutionary[4,5] and usage costs[6] for the host (reviewed in ref. 7), and infection may lead to the re-allocation of resources from other life history traits into immunity[8,9]. Immune responses can also lead to immunopathology[10,11]. Therefore, whether a pathogen persists not only depends on how well the pathogen can survive and replicate inside the host, but it also depends upon the costs of infection *versus* the costs and effectiveness of the immune response against the infection.

Persistent infections can have important consequences, e.g., prolonged host effort to clear the infection, accumulation of infection costs for the host, increased chance for within-host pathogen evolution, and increased pathogen transmission in the host population.

These phenomena could also have wide-reaching effects given the broad range of taxa found to sustain persistent bacterial infections, e.g., humans[12], other vertebrates[13–16], and insects[17]. After bacterial injection, insects have been shown to sustain systemic infections[17–20] that persist for at least 28 days[17,21], although longer term estimates are lacking. Disparate bacterial species have been shown to persist for at least seven days inside the host species used in this study, *D. melanogaster*[19,20,22–27]. At the other end of the spectrum, bacterial infections can be cleared. This infection outcome has not commonly been reported in insects (but see e.g. refs. 25, 28), and we have a limited understanding of the relationship between the different infection outcomes, i.e., fatal infection, persistent infection or clearance of infection, especially in insects.

Fatal infection outcomes are linked to pathogen virulence, which is defined as the decrease in host fitness caused by a pathogen[29]. Here, we measure virulence as reduced host survival rate. Virulence will be

[1]Institute of Biology, Freie Universität Berlin, Berlin, Germany. [2]Institute of Biology, University of Neuchâtel, Neuchâtel, Switzerland. [3]Institute of Integrative Biology, ETH Zurich, Zurich, Switzerland. [4]These authors contributed equally: Beatriz Acuña Hidalgo, Luís M. Silva, Mathias Franz. ✉e-mail: sophie.armitage@fu-berlin.de

influenced by host and pathogen traits. From the pathogen perspective, variation in virulence across pathogen strains could be due to differences in exploitation, where an increase in virulence is caused by an increase in pathogen load[30,31] (Fig. 1a). However, variation in virulence could also be due to differences in per parasite pathogenicity (PPP), i.e., the damage inflicted by each individual pathogen[30,31] (Fig. 1b). Exploitation and PPP are conceptually and mechanistically distinct, and both are important to consider to fully understand the ways in which pathogens harm their hosts. Here we hypothesised that the differences in the nature of exploitation and PPP could also differentially affect the rate of pathogen clearance.

To derive testable predictions of how exploitation and PPP could affect the rate of pathogen clearance, we developed a conceptual framework that is based on a fundamental trade-off faced by the host (Fig. 2 and Supplementary Note). This trade-off relates to the strength of the resistance mechanisms that act towards pathogen clearance, which we conceptualise here as host clearance effort. A higher host clearance effort is beneficial to the host as it increases the rate with which a pathogen is cleared (Fig. 2a), which in turn increases host survival (Fig. 2b). However, a higher clearance effort is also associated with higher immune defence costs, e.g., due to resource depletion[32], increased metabolic rate and loss of body mass[33–36] or immunopathology resulting from autoreactive self-damage[37–39] (Fig. 2c). Increasing immunity costs in turn imply that host survival decreases with increasing host clearance effort (Fig. 2d). The combination of survival benefits and costs leads to a hump-shaped relationship between host clearance effort and host survival (Fig. 2e), which implies that there is an optimal host clearance effort that maximises host survival.

To connect host clearance effort and the related clearance rate to exploitation and PPP we here extend the framework proposed by Råberg & Stjernman[31] (Fig. 2f; see also Supplementary Fig. 1 for a framework that explicitly includes the effects of host tolerance and resistance). In our framework (Fig. 2f) we consider how pathogen traits (PPP and exploitation) and a host trait (clearance effort) jointly contribute to the emergence of virulence and clearance rate, which together determine the likelihood of three fundamental infection outcomes: fatal infection, persistence infection and clearance of infection.

Based on this framework we derived three hypotheses for how exploitation and PPP could affect the process of pathogen clearance. The first hypothesis (H1) focusses on PPP and is based on the idea that increased PPP lowers host survival. Therefore, a higher PPP leads to lower host survival for a given host clearance effort (Fig. 2h). Accordingly, changes in PPP change the benefits of host clearance effort (Fig. 2a, b) without, however, changing its costs (Fig. 2c, d). As a result, we hypothesise that the optimal host clearance effort (Fig. 2e) increases with increasing PPP (Fig. 2i). In short, if a pathogen is deadlier, the host benefits from an increased host clearance effort because the increased benefits outweigh the increased immunity costs. Because of the hypothesised change in host clearance effort, we derived the prediction (P1) that the clearance rate increases with increasing PPP (Fig. 2j).

For exploitation we propose two different hypotheses (H2 and H3), which result in opposing predictions. First, we reasoned that similar to PPP, increasing exploitation increases virulence. Accordingly, our first hypothesis for exploitation (H2) and the related prediction (P2) are the same as the hypothesis and prediction for PPP: host clearance effort increases with increasing exploitation, which in turn leads to an increased clearance rate (Fig. 2g–j). In contrast, our last hypothesis (H3) is based on the idea that in addition to affecting virulence (Fig. 2g–j), higher exploitation should make it harder for the host to clear the infection. Thus, for a given host clearance effort the clearance rate should be lower for a higher exploitation (Fig. 2k). Accordingly, higher exploitation should also result in reduced host survival (Fig. 2l). However, reduced survival benefits in this case do not necessarily affect the optimal host clearance effort (Fig. 2m). Nevertheless, due to its direct influence on clearance rate, we can derive the prediction (P3) that increased exploitation results in a lower clearance rate (Fig. 2n).

In this study we experimentally infected *Drosophila melanogaster* to investigate variation in infection outcomes, i.e., fatal, persistent and cleared infections. For this purpose, we used mainly bacterial species previously isolated from wild flies, which were chosen based on studies that together suggest they may show a range of virulence[23–25,40–42]. We assessed variation in virulence and persistence across a range of infection doses. We then used the concepts of exploitation and PPP to disentangle the causes of variation in virulence across species.

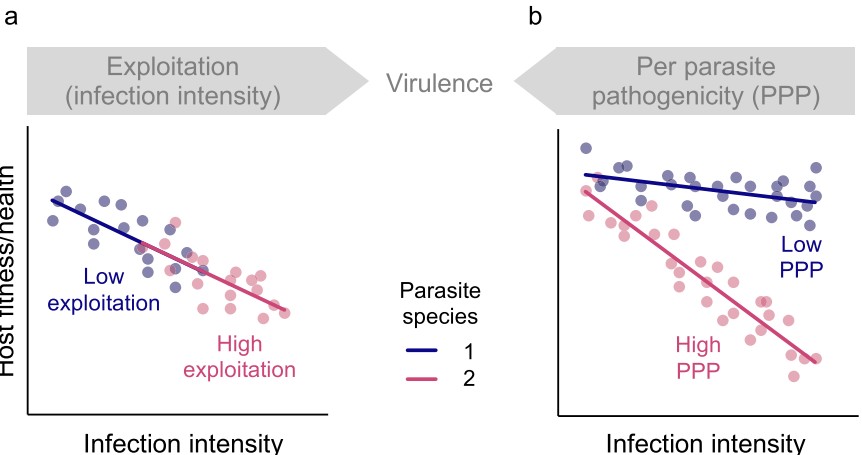

**Fig. 1 | Decomposing virulence.** Both exploitation and per parasite pathogenicity (PPP) can harm the host and thereby contribute towards virulence. Exploitation describes the infection intensity, or parasite/pathogen load, inside the hos PPP describes the damage per parasite that an infection does to the host. Here there is variation among parasite species in exploitation and PPP, as illustrated by hypothetical relationships between host fitness and infection intensity for two species of parasite infecting the same host genetic background. **a** The parasite species have the same PPP but differ in exploitation. Parasite 1 has lower exploitation compared with parasite 2, because it causes a lower infection intensity. **b** In contrast with **a** the parasite species have the same average exploitation but species 1 has lower PPP because its reaction norm has a shallower slope. This means that compared with species 2, species 1 causes less damage to the host with increasing parasite load. Figure modified from Råberg[30].

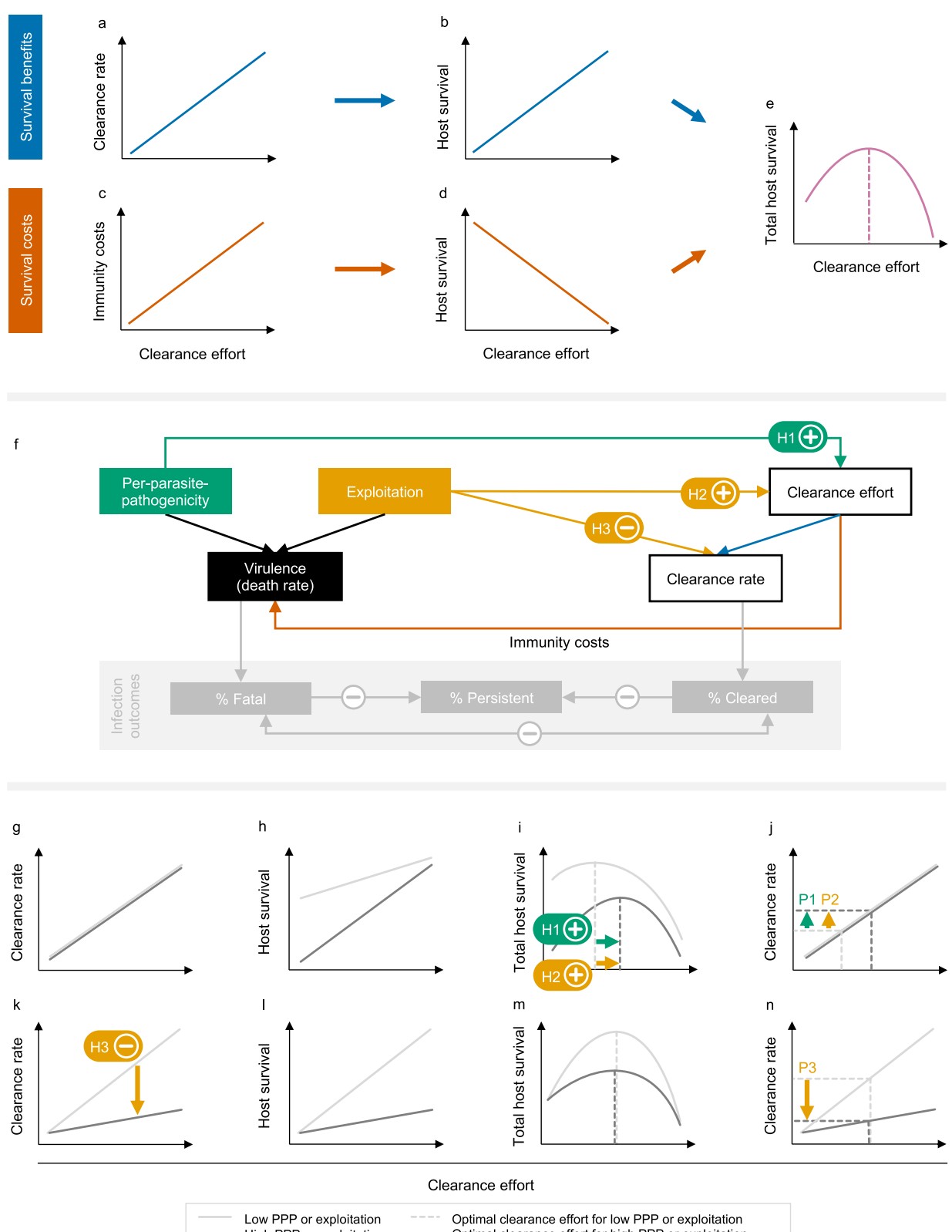

Finally, we used these results to test our predictions about how exploitation and PPP affect clearance rates. We find that virulence differences across bacteria species can be explained by variation in exploitation and PPP. Furthermore, when there is higher exploitation early in the infection, clearance rates later in the infection are lower (P3). However, there is no apparent effect of PPP on clearance rates. We explain these results in the context of a novel conceptual framework. Decomposing virulence into its constituent parts is broadly applicable and can lend useful insight into infection processes.

## Results

### Bacterial species vary in virulence

We injected female flies with one of three bacterial species that had been isolated from wild-collected *D. melanogaster*: *Lactococcus lactis*[43],

**Fig. 2 | Framework and hypotheses for the relationship between per parasite pathogenicity (PPP), exploitation, virulence and clearance rate. a–e** Host efforts to clear the pathogen generate costs and benefits via changes in **a** clearance rate and **c** immunity costs. **b, d** Clearance rate and immunity costs affect host survival and, **e**, in combination lead to an optimal host clearance effort (vertical dotted line) that maximises host survival. Host clearance effort is conceptualised as the strength of the resistance mechanisms that act towards pathogen clearance. **f** Schematic for how PPP, exploitation, and host clearance effort affect virulence and clearance rate: combined, these determine the three infection outcomes. This scheme extends the framework proposed by Råberg & Stjernman[31], which proposes that PPP and exploitation are different determinants of virulence. The green and yellow labelled arrows represent our hypotheses and predictions. **g–j** Illustration of hypotheses H1 and H2, which are based on the idea that PPP and exploitation can each increase virulence. Two scenarios are depicted: low PPP or exploitation, and high PPP or exploitation (see legend). **g** The relationship between host clearance effort and clearance rate is not affected by high/low PPP or exploitation. **h** However, increased PPP or exploitation lower host survival. **i** These differential survival benefits of host clearance effort lead to the hypotheses that increases in PPP (H1) and exploitation (H2) lead to an increase in the optimal host clearance effort. **j** As a result, increased clearance rates are predicted with increasing PPP and exploitation (predictions P1 and P2 respectively). **k–n** Illustration of hypothesis H3, which is based on the idea that in addition to affecting virulence (**g–j**), higher exploitation makes it harder for the host to clear the infection. **k** Thus, for a given host clearance effort the clearance rate should be lower for higher exploitation, which results in lower host survival (**l**). **m** When considering total host survival, these differential survival benefits do not necessarily affect the optimal host clearance effort. **n** However, due to its direct influence on clearance rate, higher exploitation is predicted to lead to a lower clearance rate (P3). See Supplementary Note.

*Providencia burhodogranariea*[44] and *Pseudomonas entomophila*[45], and one species that has been detected in the microbiota of *D. melanogaster*[46] and was isolated from a maize plant: *Enterobacter cloacae*. Fly survival after injection with five doses of the four different bacterial species is shown in Fig. 3a–d. As predicted, the bacterial species differed significantly in virulence, given as the maximum hazard ($F_{3,55} = 193.05$, $p < 0.0001$; Fig. 3e; Supplementary Tables 1 and 2). In all further figures, the bacterial species are therefore ordered by virulence, as shown in Fig. 3e.

## Differences in virulence are due to variation in exploitation and PPP

Simultaneously to testing survival after infection, in a separate cohort of flies, we assessed infection intensity over time post injection (Fig. 4). From this data, we used the geometric mean of the bacterial load from the first two days post injection, as a proxy for pathogen exploitation (see methods for rationale). *Ps. entomophila* was not included in this analysis because the maximum hazard was consistently reached at around day one post injection. The bacterial species varied significantly in host exploitation during this early infection phase ($F_{2,35} = 35.87$; $p < 0.0001$; Fig. 5a), where the order of exploitation among the three species followed the same order of virulence as reported above.

The slopes of the relationship between infection intensity and maximum hazard differed significantly across bacterial species, suggesting that the bacterial species also differed in their PPP (infection intensity × bacterial species: $F_{2,39} = 7.35$, $p = 0.0020$; Fig. 5b). *E. cloacae* had a relatively flat reaction norm, indicating a minimal increase in hazard with an increase in bacterial load, and therefore a significantly lower PPP compared to both *Pr. burhodogranariea* (Tukey contrast: $t = -3.74$; $p = 0.0017$) and *L. lactis* ($t = -3.34$; $p = 0.0052$). In contrast, the latter two species had similar PPP to each other ($t = -0.68$; $p = 0.78$): both species had negative reaction norms, indicating an increase in hazard with an increase in bacterial load. Qualitatively similar results were obtained using three different values to estimate the maximum hazard (Supplementary Fig. 2). The underestimate of the load of some of the *L. lactis*-injected flies (see methods) is unlikely to have affected the interpretations of the data in Fig. 5a: a higher load would have acted to exacerbate the higher load for *L. lactis*. For Fig. 5b a higher load would have dragged the *L. lactis* data points further to the right so that it may have flattened the slope.

## All bacterial species established persistent infections

To date, experimentally-induced septic infections have been found to persist for at least 28 days in the insects *Tenebrio molitor*[17] and *D. melanogaster*[24]. In the chronic infection phase, the bacterial load can stabilise around a relatively constant pathogen load[20,25], termed the set

point bacterial load (SPBL; ref. 25), after the set point viral load (e.g. ref. 47). By homogenising living flies, we found that the two species with lower virulence, *E. cloacae* and *Pr. burhodogranariea* were able to persist inside the fly until at least 35 days post injection (Fig. 4a and b respectively). The persistence estimates for *L. lactis* (28 days; Fig. 4c) and *Ps. entomophila* (four days; Fig. 4d) were both shorter, because the high mortality caused by these bacterial species meant that we could not test later time points. To uncover whether infections can persist for even longer, we also tested for the presence or absence of bacteria in flies that had died between 14 and 35 days and between 56- and 78-days post injection. These dead flies originated from daily survival checks made during the experiment to estimate virulence. We indeed found that infections could persist for considerably longer than previous estimates, i.e., around two and a half months: *E. cloacae* = 77 days, *Pr. burhodogranariea* = 78 days, *L. lactis* = 76 days and *Ps. entomophila* = 75 days (Supplementary Fig. 3a–d).

## Injection dose correlates with persistent infection loads

Previous studies have found a positive relationship between the initial injection dose and the SPBL, at seven- and fourteen-days post injection[22,25], and it has been suggested that the SPBL will remain at around the bacterial load at which the infection was controlled[22,25]. We tested this relationship at seven days post injection, finding that *E. cloacae* (Supplementary Fig. 4a) and *Pr. burhodogranariea* (Supplementary Fig. 4b) loads were significantly positively correlated with the initial injection dose (Supplementary Table 5). *L. lactis* loads showed no significant correlation with the initial injection dose (Supplementary Fig. 4c; Supplementary Table 5), and we hypothesised that this might be due to our underestimation of the load of some flies injected with this bacterial species (see methods). When we excluded the two flies at day seven that had been assigned the maximum value, the relationship became statistically significant (injection dose: $F_{1,35} = 4.59$; $p = 0.039$; Supplementary Table 5).

## Clearance of infections

Summing up across doses and days, 39.4% (177 of 449) of *E. cloacae*-injected flies, 11.8% (45 of 381) of *Pr. burhodogranariea*-injected flies, 3.7% (11 of 301) of *L. lactis*-injected flies, and 21.4% (15 of 70) of *Ps. entomophila*-injected flies cleared the infections (see Methods for details) (Fig. 6a–d).

## Clearance rate varies according to virulence

We tested whether there was variation in clearance rate of the different bacterial species. For this we used the latest time frame for which we could calculate this index for all four species, which was days three and four post injection. The four species covered a broad spectrum of clearance rates (Fig. 7a). There was a statistically significant difference among species (Chisq = 15.31, df = 3, $p = 0.0016$). After $p$-value correction, there were statistically significant differences among the

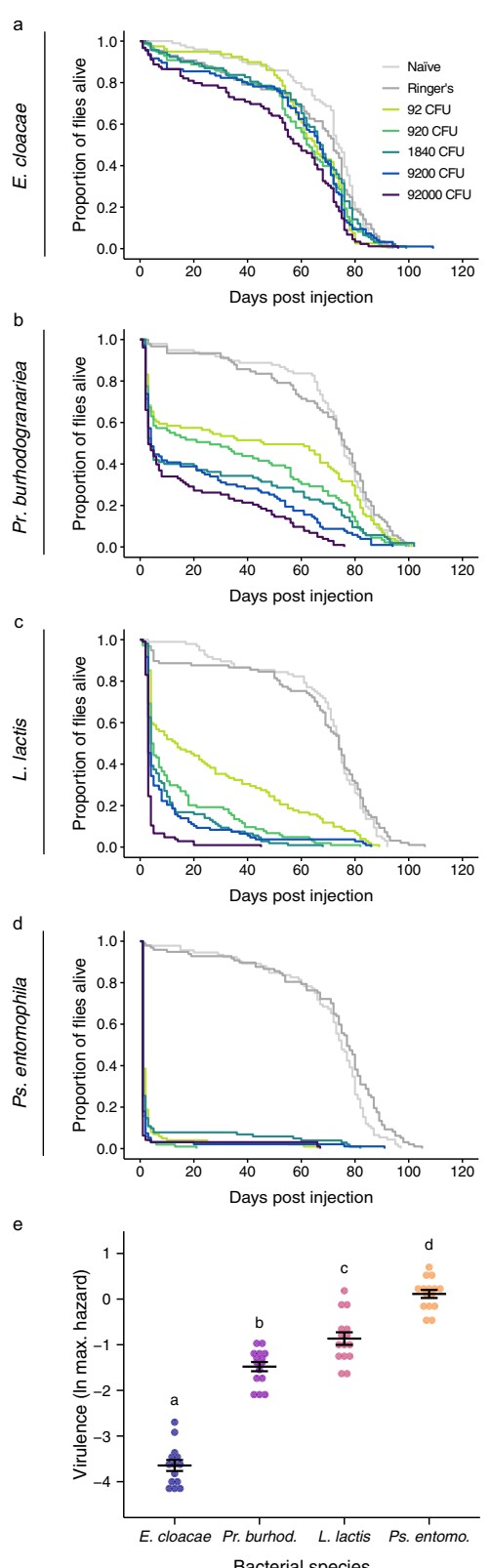

**Fig. 3 | Fly survival and virulence after injection with one of four bacterial species. a–d** Survival curves after injection with four bacterial species, each at one of five doses. Controls were injected with Ringer's solution or received no injection (naïve). The legend in panel a shows the treatments for all survival curves, where CFU denotes colony forming units. Flies were housed in vials in groups of six. In the following, *n* = the number of flies examined over three independent experiments and n is given in the order of the treatments as shown in the legend: *Enterobacter cloacae* n = 99, 86, 79, 98, 96, 89, 92; *Providencia burhodogranariea* n = 98, 91, 101, 89, 103, 103, 105; *Lactococcus lactis* n = 96, 97, 102, 104, 108, 106, 107; *Pseudomonas entomophila* n = 92, 97, 101, 102, 95, 96, 102. **e** Virulence measured as the natural log of maximum hazard for all bacterial species, where each data point is the maximum hazard between zero- and 20-days post injection, calculated from one experimental replicate per bacterial dose. In the following, *n* = the number of experimental replicates/doses per bacteria: *E. cloacae* n = 14, all other bacteria n = 15. Ringer's injected and naïve flies are not included. Black lines show means and standard errors. We fitted a linear model (two-tailed) where the main effect of bacterial species was *p* < 0.0001 (see main text). See Supplementary Table 1 for multiple comparisons and effect sizes. Different letters denote means that are significantly different from one another.

higher levels of clearance rates compared to the two species of intermediate virulence (Fig. 7a).

## Exploitation but not PPP predicts clearance rate

We next used the virulence decomposition framework to analyse how the components of virulence relate to variation in clearance rate. We found no support for our first hypothesis (H1), and its derived prediction that clearance rate increases with increasing PPP (Fig. 2i), as we could not detect any statistically significant effect of PPP on clearance rate, using clearance index$_{3,4}$ and clearance index$_{7,14,21}$ (Supplementary Table 6). However, there was a significant negative effect of exploitation, showing that as the mean bacterial load increased, clearance rate later in the infection decreased (Fig. 7b, c, Supplementary Table 6). This finding is inconsistent with prediction P2 of a positive relationship (Fig. 2j), i.e., that clearance rate increases with increasing exploitation, and it is instead consistent with prediction P3, i.e., that increased exploitation results in a lower clearance rate (Fig. 2n). Accordingly, this finding supports our third hypothesis (H3). Similar results were obtained using the three alternative smoothing parameters for calculating PPP (Supplementary Fig. 5). The underestimate of the load of some of the *L. lactis*-injected flies is unlikely to have affected the interpretations of Fig. 7b, c: a higher load would have dragged the *L. lactis* data points further to the right, giving a longer tail to the distribution.

## Bacterial clearance rate before death is dose dependent

We also analysed bacterial clearance rate in dead flies. For all bacterial species there were flies where the infection persisted until death, and flies that were uninfected at death (Supplementary Fig. 3a–d). Lower injection doses of *E. cloacae* and *Pr. burhodogranariea* were more likely to be cleared before death than higher injection doses (Supplementary Fig. 3e, f respectively; Supplementary Table 7; see also Supplementary Fig. 6). *L. lactis* and *Ps. entomophila* could not be tested due to low sample sizes.

## Discussion

We have shown that sustained persistent infection and clearance are both possible outcomes for bacteria causing a range of virulence when they infect female *D. melanogaster*. Bacteria of all species can persist inside the host for at least 75 days and bacterial virulence differences can be explained by a combination of variation in exploitation and PPP. We have shown that clearance rates are bacteria species specific, and in a novel application of decomposing virulence, we found that exploitation and PPP have different effects on clearance rates.

following species pairs: *E. cloacae* and *Pr. burhodogranariea* (*p* = 0.024), *E. cloacae* and *L. lactis* (*p* = 0.011), *Ps. entomophila* and *Pr. burhodogranariea* (*p* = 0.048), *Ps. entomophila* and *L. lactis* (*p* = 0.011). Rather than matching the virulence gradient across species (Fig. 3e), clearance rates formed a U-shaped pattern with the species with the highest virulence (*Ps. entomophila*) and lowest virulence (*E. cloacae*) showing

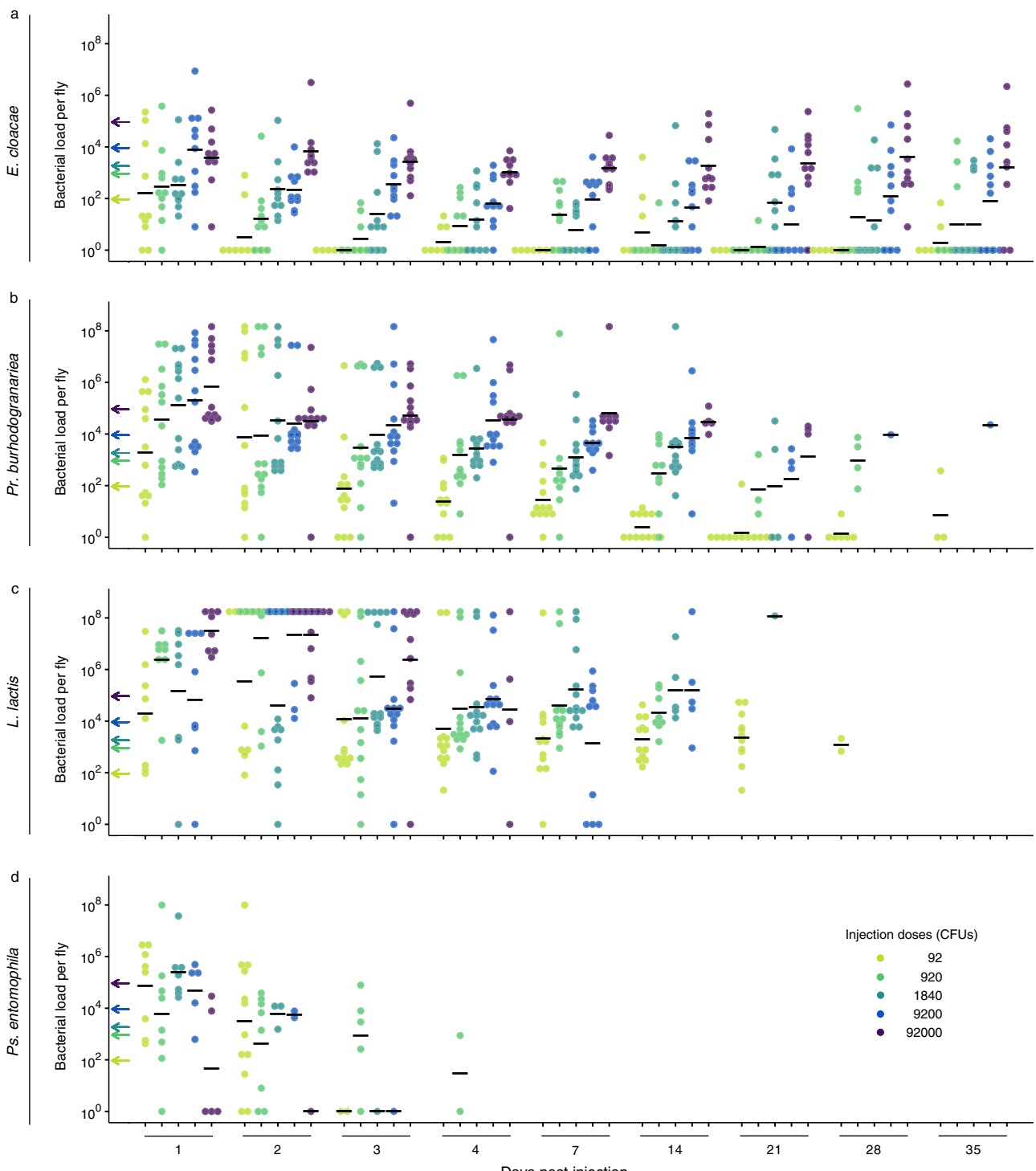

**Fig. 4 | Bacterial load per living fly after injection with one of four bacterial species. a–d** Flies were injected with either *Enterobacter* cloacae, *Providencia burhodogranariea*, *Lactococcus lactis* or *Pseudomonas entomophila* and then homogenised at between 1- and 35-days post injection. The injection dose legend for all panels is shown in **d** where CFU denotes colony forming units. The arrows on the y-axis indicate the approximate injection doses. Each data point is the bacterial load of one fly. Reducing data points on the x-axis are due to increasing fly death over time. Sample sizes can be found in Supplementary Table 3a–d. Black lines show geometric means.

## Decomposing virulence

To understand why the infecting species showed pronounced differences in virulence, we decomposed virulence into its two components: exploitation and PPP[30,31]. Exploitation, given as infection intensity, is the more frequently tested explanation for variation in virulence[31]. There is evidence that exploitation varies across parasite genotypes[48–50], and also, unsurprisingly, that it varies across parasite species infecting the same host genotype[22,23,25]. We found that the bacterial species varied significantly in exploitation, whereby bacterial load increased as virulence increased.

However, quantifying the harm or damage caused per parasite, i.e., PPP[30,31], is essential to have a full picture of how the pathogen impacts its host. *Pr. burhodogranariea* and *L. lactis* had higher PPP compared to *E. cloacae*. Combined with the exploitation results, this

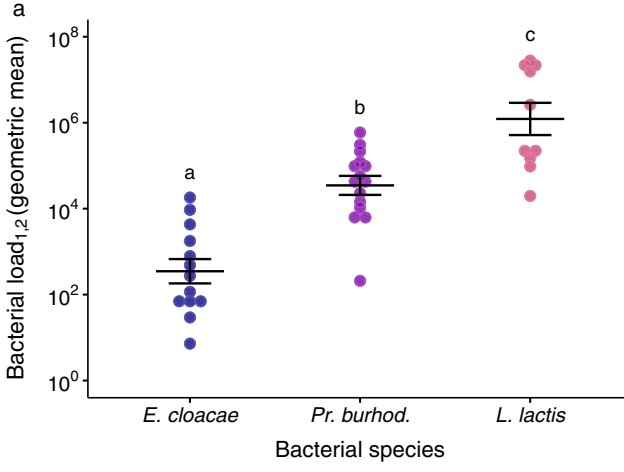

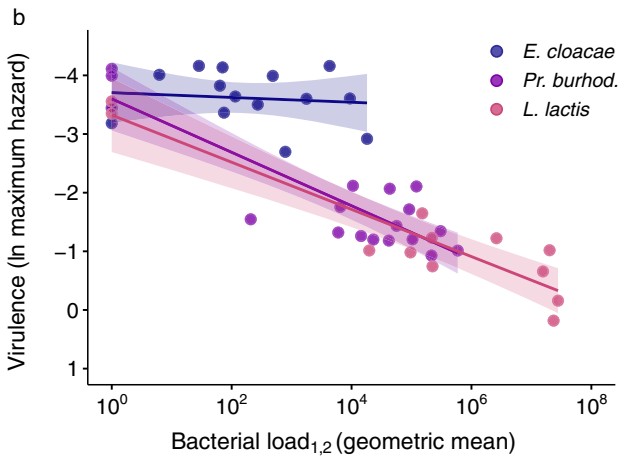

**Fig. 5 | Virulence decomposition. a** Pathogen exploitation given as infection intensity/bacterial load across species. Each data point is one of five injection doses per bacterial species, per experimental replicate, and gives the geometric mean of bacterial load for days one and two post injection (denoted as $_{1,2}$), where *Enterobacter cloacae* $n = 13$, *Providencia burhodogranariea* = 15, *Lactococcus lactis* = 10. The circles are jittered along the x-axis to aid visualisation of overlapping data points. Black lines show means and standard errors. We fitted a linear model (two-tailed) where the main effect of bacterial species was $p < 0.0001$ (see main text). Different letters denote means that are significantly different from one another. See Supplementary Table 4 for multiple comparisons and effect sizes. **b** PPP given as the relationship between bacterial load and the inverse of maximum hazard, so that the virulence increases with proximity to the x-axis. The bacterial load data is the same as that given in **a** but with the addition of the Ringer's control group, giving the following total sample sizes *Enterobacter cloacae* $n = 15$, *Providencia burhodogranariea* = 18, *Lactococcus lactis* = 12. To allow inclusion of the uninfected Ringer's control group to the figure, we added one CFU to all mean bacterial load values. The natural log of maximum hazard data is estimated from survival data for the corresponding injection doses and experimental replicates (as in Fig. 3e). Lines show linear regressions with 95 % confidence intervals.

implies that *E. cloacae* is less virulent because of both lower PPP and less exploitation. Furthermore, given that *Pr. burhodogranariea* and *L. lactis* showed similar levels of PPP, it suggests that the difference in virulence between these two species is due to higher exploitation by *L. lactis*, rather than differences in PPP. Exploitation and PPP have not frequently been utilised in the same study (but see refs. 31, 48), yet the decomposition framework can provide highly valuable insight into infection processes, such as the fact that PPP explains more of the variance in virulence across HIV-1 genotypes compared to the more often measured set-point viral load (measure of exploitation)[51].

## Bacterial persistence
All four bacterial species were able to establish persistent infections in *D. melanogaster* for at least 75 days, which is far beyond the currently known length of persistent infections after injection in insects (28 days: refs. 17, 24). The duration of infection can be of key ecological, and potentially also evolutionary, importance because persistence determines the prevalence of infection in a population, and therefore could affect transmission. It is unclear how these bacterial species are able to persist for so long inside the host, although there are a number of theoretical possibilities, for example, through surviving inside host tissue, forming biofilms or existing as persister or tolerant cells[52].

*D. melanogaster* that were able to control a *Providencia rettgeri* infection during the acute infection phase had a relatively constant bacterial load until at least ten days post injection[25]. Our bacterial load data (Fig. 4) lends support to the set point bacterial load (SPBL) concept, and to the idea that virulence relates to SPBL[25], given that low virulence *E. cloacae* had a persistent load of tens to hundreds of bacteria, and high virulence *L. lactis* had a load of tens of thousands of bacteria (Fig. 4). The relationship between virulence and bacterial load is statistically supported by our virulence decomposition analysis, which shows that as virulence increases, so does exploitation of the host over the first couple of days post infection. Therefore, more virulent bacteria have higher initial proliferation rates as shown by exploitation. Given that the infection load stays relatively constant in the longer term, the initial proliferation differences likely explain the relationship between SPBL and virulence.

The initial injection dose of *E. cloacae* and *Pr. burhodogranariea* correlated positively with the bacterial load at seven days post injection. Our results expand the known bacterial species for which this relationship exists, and they lend weight to the idea that this may be a more general phenomenon in *D. melanogaster* bacterial infections. Given that insects can show dose dependent inducible immune activation[53], and that the antimicrobial peptide (AMP) Drosocin has been shown to control *E. cloacae* infections and that a combination of AMPs control *Pr. burhodogranariea* infections[42], one could hypothesise that these AMPs are to some degree involved. However, the mechanisms that allow a dose-dependent persistent infection, remain to be uncovered.

## Clearance and virulence
Our finding that the low virulent species could be cleared, is supported by evidence from other bacterial species[25]. However, our data challenge the finding that clearance in intermediate and high virulence pathogens is rare[25] because the three more virulent bacteria all appear to be clearable to differing degrees. Persistent infections are therefore not inevitable. We expected that there may be selection for a fast and efficient early clearance of infection by *Ps. entomophila*, because of its high virulence. The clearance rate of *Ps. entomophila* was indeed higher than for the intermediate bacteria, although mortality was too high to assess clearance rates in living flies for longer than four days post injection. Nonetheless, there is evidence from other studies that *Ps. entomophila* has high virulence and can be cleared from other *D. melanogaster* populations/genotypes[24,41,54]. Five out of ten tested *D. melanogaster* genotypes contained some individuals who cleared *Ps. entomophila*[41]; host genotypic variation in clearance rates may thereby more generally explain why some studies observe clearance and others do not. The patterns of bacterial clearance in dead homogenised flies largely reflected the results for live flies.

## Clearance and decomposed virulence
We next sought to understand how the two components that determine variation in virulence affect clearance of the infection. The results seem to suggest that exploitation and PPP can affect clearance rates in different ways, which could result in different, and potentially even

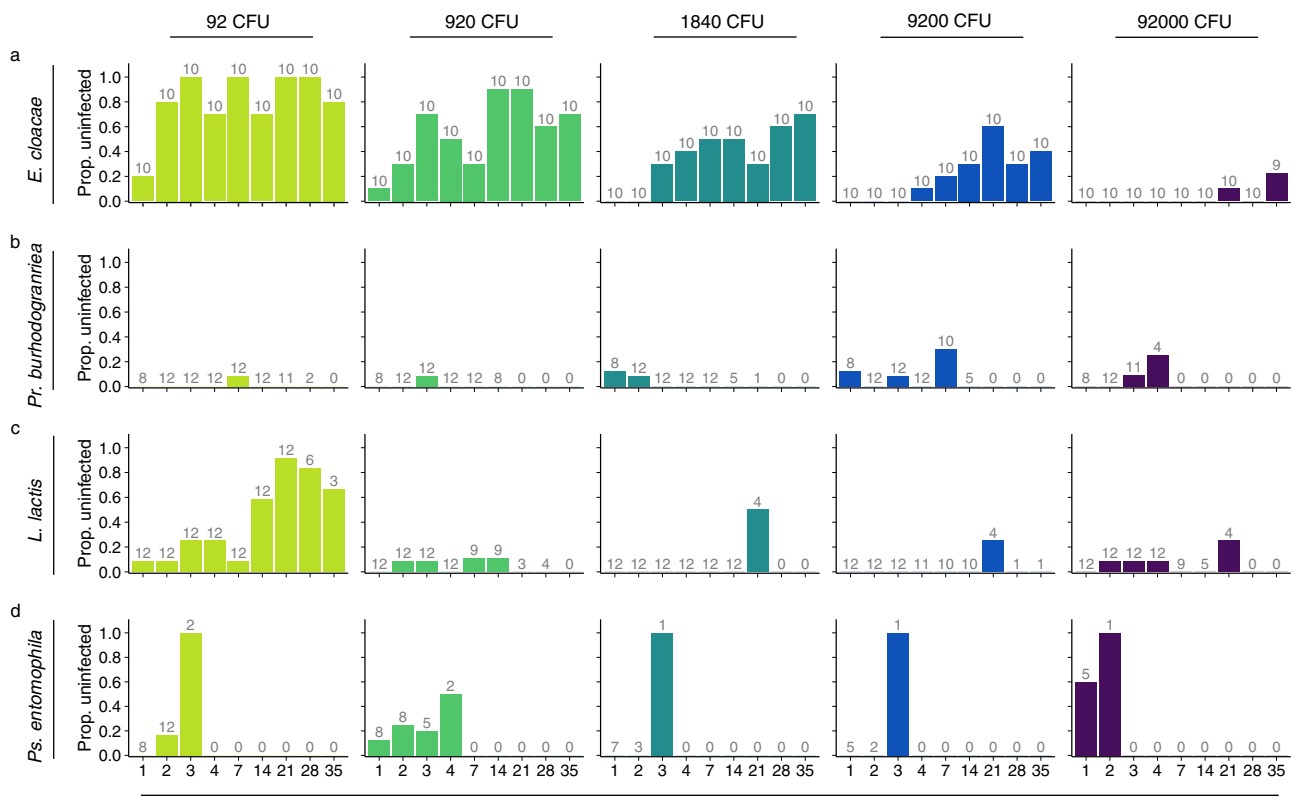

**Fig. 6 | Bacterial clearance by living flies.** The proportion of live flies that were found to be uninfected (for bacteria detection limit see Methods) at different times post injection with **a** *Enterobacter* cloacae, **b** *Providencia burhodogranariea*, **c** *Lactococcus lactis* or **d** *Pseudomonas entomophila*. Each column shows a different injection dose. Numbers above the bars indicate the total numbers of flies from which the proportions were calculated, i.e., the total numbers of flies homogenised. Zeros on the x-axis mean that there were no flies alive from which to assess clearance.

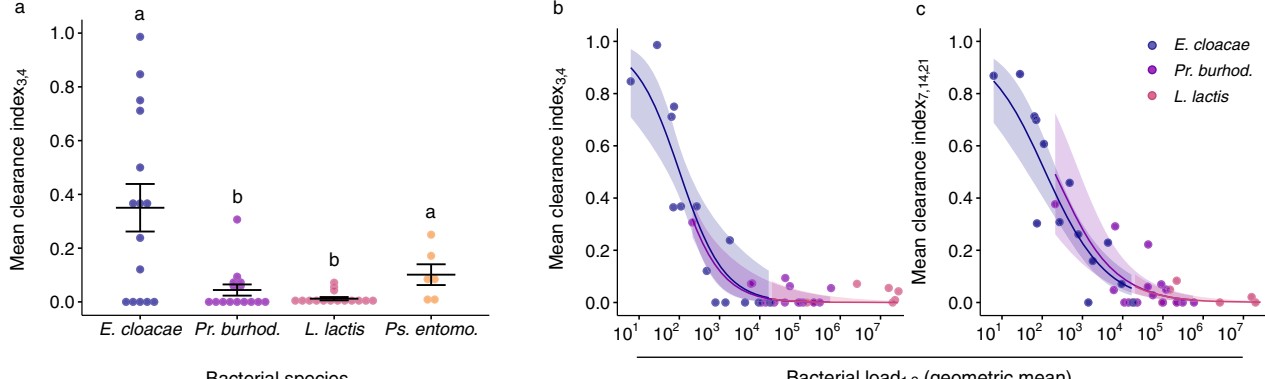

**Fig. 7 | Effect of bacterial species and exploitation on clearance rate.** For all figures, each data point is from one injection dose per bacteria, per experimental replicate, and gives the mean proportion of cleared infections (out of the initial infected population) on days three and four (clearance index$_{3,4}$) or days seven, 14 and 21 (clearance index$_{7,14,21}$). **a** Mean species differences in clearance index$_{3,4}$. For *Pseudomonas entomophila* $n = 6$, for *Enterobacter cloacae*, *Providencia burhodogranariea* and *Lactococcus lactis* $n = 15$. The circles are jittered along the x-axis to aid visualisation of overlapping data points. Black lines show means and standard errors. We fitted a linear model (two-tailed) where the main effect of bacterial species was $p = 0.0016$ (see main text). Different letters denote means that are significantly different from one another (Mann-Whitney-U post hoc tests, two-tailed tests, see main text for statistical results). The effect of pathogen exploitation, given as bacterial load, upon **b** mean clearance index$_{3,4}$ and **c** mean clearance index$_{7,14,21}$. The geometric mean of bacterial load was calculated from days 1 and 2 post injection (denoted as $_{1,2}$), i.e., the same values as in Fig. 4. The negative relationships between the two variables are shown with 95 % confidence intervals. Statistics are given in the main text and the species legend for both panels is shown in **c**. The negative relationships in **b** and **c** support H3 and P3 (Fig. 2k, n).

opposing, patterns of how variation in virulence is related to clearance rates. In agreement with our third prediction (P3), we found that across species, increased host exploitation early in the infection, i.e., days one and two post-infection, was associated with decreased clearance rates at days three and four post-infection, and at days seven to 21. This finding is consistent with our hypothesis H3 that changes in exploitation do not necessarily change the effort that the host invests in clearance, and instead affect how much clearance can be achieved for a

given host clearance effort (Fig. 2k–n). The negative relationship between exploitation and clearance rate contrasts with data from vertebrate viral infections (reviewed in ref. 55, e.g. ref. 56), where larger viral loads led to a faster decline in viral load or in shorter durations of viremia. Although these studies did not directly assess clearance rates as defined in our study, their results are consistent with our hypothesis H2, where increasing exploitation leads to increased optimal clearance effort, which in turn leads to increased clearance rate (Fig. 2g–j).

We did not find support for our prediction P1 that increasing PPP would be associated with an increased clearance rate, which is expected if PPP alters the benefits of host clearance effort and accordingly the optimal effort that the host should invest in clearing the pathogen (Fig. 2g–j). Although we note that the statistical power of detecting an effect of PPP on clearance rate was lower than the power to detect an effect of exploitation. *Ps. entomophila* produces a virulence factor[57], and a pore-forming toxin called Monalysin[58] in association with the activation of stress-induced pathways and an increase in oxidative stress[59] leading to a lack of tissue repair in the gut after oral infections. If similar pathologies are induced in the haemocoel after infection, clearance of this species (Fig. 7a) might have been caused by particularly high levels of PPP. In contrast to other bacterial species, sustaining a persistent bacterial load in the face of high levels of tissue damage might rarely be a viable option. Taken together, our results suggest that for the investigated pathogens, variation in clearance rates does not emerge from pronounced variation in host clearance effort as expected for hypotheses H1 and H2. Instead, the observed variation in clearance rates is best explained by variation in the load-related difficulty to clear infections (as proposed by hypothesis H3).

The conceptual framework that we present here can be used to formulate some basic predictions about immunity. For example, we proposed that an increased clearance rate is mechanistically mediated by an increased host clearance effort (H1 and H2), and we could predict that this increased clearance effort would be due to a stronger immune response. In contrast, H3 is not mediated by a change in host clearance effort – and thus is not predicted to result in a change in immune response. We also note that these hypotheses are not necessarily mutually exclusive. However, the framework (necessarily) does not attempt to take the full complexities of host-pathogen interactions into account. For example, bacterial infections are dynamic processes, where bacterial load (e.g. ref. 25) and the host immune response towards the pathogen can change over time, and where they vary according to the infecting microorganism[60]. These parameters will also likely vary depending upon the host-pathogen interaction being considered.

In conclusion, our results strongly support that PPP is an important component driving variation in virulence, and that disentangling its contribution towards virulence, in combination with the contribution of exploitation, will undoubtedly help our mechanistic and evolutionary understanding of host-pathogen interactions. Our framework provides a general conceptual contribution to the field and as such could be applicable across a broad range of host-pathogen interactions. We suggest that such a decomposition of virulence can be used to better understand how virulence relates to other infection processes such as infection clearance and the effort that goes into clearance, and it may also contribute towards expanding the theory on virulence evolution. Furthermore, when designing drugs in the medical and veterinary sciences, a better understanding of the contributions of PPP and exploitation towards virulence, may help to predict the evolutionary implications of a treatment.

## Methods
### Fly population and maintenance
We used an outbred population of *Drosophila melanogaster* established from 160 *Wolbachia*-infected fertilised females collected in Azeitão, Portugal[54], and given to us by Élio Sucena. For at least 13

generations prior to the start of the experiments the flies were maintained on standard sugar yeast agar medium (SYA medium: 970 ml water, 100 g brewer's yeast, 50 g sugar, 15 g agar, 30 ml 10% Nipagin solution and 3 ml propionic acid; ref. 61), in a population cage containing at least 5000 flies, with non-overlapping generations of 15 days. They were maintained at $24.3 \pm 0.2\,°C$, on a 12:12 h light-dark cycle, at 60–80 % relative humidity. The experimental flies were kept under the same conditions. No ethical approval or guidance is required for experiments with *D. melanogaster*.

### Bacterial species
We used the Gram positive *Lactococcus lactis* (gift from Brian Lazzaro), Gram negative *Enterobacter cloacae subsp. dissolvens* (hereafter called *E. cloacae*; German collection of microorganisms and cell cultures, DSMZ; type strain: DSM-16657), *Providencia burhodogranariea* strain B (gift from Brian Lazzaro, DSMZ; type strain: DSM-19968) and *Pseudomonas entomophila* (gift from Bruno Lemaitre). *L. lactis*[43], *Pr. burhodogranariea*[44] and *Ps. entomophila*[45] were isolated from wild-collected *D. melanogaster* and can be considered as opportunistic pathogens. *E. cloacae* was isolated from a maize plant, but has been detected in the microbiota of *D. melanogaster*[46]. All bacterial species were stored in 34.4% glycerol at $-80\,°C$ and new cultures were grown freshly for each experimental replicate.

### Experimental design
For each bacterial species, flies were exposed to one of seven treatments: no injection (naïve), injection with *Drosophila* Ringer's (injection control) or injection with one of five concentrations of bacteria ranging from $5 \times 10^6$ to $5 \times 10^9$ colony forming units (CFUs)/mL, corresponding to doses of approximately 92, 920, 1,840, 9200 and 92,000 CFUs per fly. The injections were done in a randomised block design by two people. Each bacterial species was tested in three independent experimental replicates. Per experimental replicate we treated 252 flies, giving a total of 756 flies per bacterium (including naïve and Ringer's injection control flies). Per experimental replicate and treatment, 36 flies were checked daily for survival until all flies were dead. A sub-set of the dead flies were homogenised upon death to test whether the infection had been cleared before death or not. To evaluate bacterial load in living flies, per experimental replicate, four of the flies were homogenised per treatment, for each of nine time points: one, two, three, four, seven, 14, 21, 28- and 35-days post-injection.

### Infection assay
Bacterial preparation was performed as in Kutzer et al.[24], except that we grew two overnight liquid cultures of bacteria per species, which were incubated overnight for approximately 15 h at $30\,°C$ and 200 rpm. The overnight cultures were centrifuged at $2880 \times g$ at $4\,°C$ for 10 min and the supernatant removed. The bacteria were washed twice in 45 mL sterile *Drosophila* Ringer's solution (182 mmol·L-1 KCl; 46 mol·L-1 NaCl; 3 mmol·L-1 CaCl2; 10 mmol·L-1 Tris·HCl; ref. 62) by centrifugation at $2880 \times g$ at $4\,°C$ for 10 min. The cultures from the two flasks were combined into a single bacterial solution and the optical density (OD) of 500 µL of the solution was measured in a Ultrospec 10 classic (Amersham) at 600 nm. The concentration of the solution was adjusted to that required for each injection dose, based on preliminary experiments where a range of ODs between 0.1 and 0.7 were serially diluted and plated to estimate the number of CFUs. Additionally, to confirm *post hoc* the concentration estimated by the OD, we serially diluted to $1:10^7$ and plated the bacterial solution three times and counted the number of CFUs.

The experimental flies were reared at constant larval density for one generation prior to the start of the experiments. Grape juice agar plates (50 g agar, 600 mL red grape juice, 42 mL Nipagin [10% w/v solution] and 1.1 L water) were smeared with a thin layer of active yeast paste and placed inside the population cage for egg

laying and removed 24 h later. The plates were incubated overnight then first instar larvae were collected and placed into plastic vials (95 × 25 mm) containing 7 ml of SYA medium. Each vial contained 100 larvae to maintain a constant density during development. One day after the start of adult eclosion, the flies were placed in fresh food vials in groups of five males and five females, after four days the females were randomly allocated to treatment groups and processed as described below.

Before injection, females were anesthetised with $CO_2$ for a maximum of five minutes and injected in the lateral side of the thorax using a fine glass capillary (Ø 0.5 mm, Drummond), pulled to a fine tip with a Narishige PC-10, and then connected to a Nanoject II™ injector (Drummond). A volume of 18.4 nL of bacterial solution, or *Drosophila* Ringer's solution as a control, was injected into each fly. Full controls, i.e., naïve flies, underwent the same procedure but without any injection. After being treated, flies were placed in groups of six into new vials containing SYA medium, and then transferred into new vials every 2–5 days. Maintaining flies in groups after infection is a standard method in experiments with *D. melanogaster* that examine survival and bacterial load (e.g. refs. [22], [63], [64]). At the end of each experimental replicate, 50 μL of the aliquots of bacteria that had been used for injections were plated on LB agar to check for potential contamination. No bacteria grew from the Ringer's solution and there was no evidence of contamination in any of the bacterial replicates. To confirm the concentration of the injected bacteria, serial dilutions were prepared and plated before and after the injections for each experimental replicate, and CFUs counted the following day.

## Bacterial load of living flies

Flies were randomly allocated to the day at which they would be homogenised. Prior to homogenisation, the flies were briefly anesthetised with $CO_2$ and removed from their vial. Each individual was placed in a 1.5 mL microcentrifuge tube containing 100 μL of pre-chilled LB media and one stainless steel bead (Ø 3 mm, Retsch) on ice. The microcentrifuge tubes were placed in a holder that had previously been chilled in the fridge at 4 °C for at least 30 min to reduce further growth of the bacteria. The holders were placed in a Retsch Mill (MM300) and the flies homogenised at a frequency of 20 Hz for 45 s. Then, the tubes were centrifuged at $420 \times g$ for one minute at 4 °C. After resuspending the solution, 80 μL of the homogenate from each fly was pipetted into a 96-well plate and then serially diluted 1:10 until 1:$10^5$. Per fly, three droplets of 5 μL of every dilution were plated onto LB agar. Our lower detection limit with this method was around seven colony-forming units per fly. We consider bacterial clearance by the host to be when no CFUs were visible in any of the droplets, although we note that clearance is indistinguishable from an infection that is below the detection limit. The plates were incubated at 28 °C and the numbers of CFUs were counted after ~20 h. Individual bacterial loads per fly were back calculated using the average of the three droplets from the lowest countable dilution in the plate, which was usually between 10 and 60 CFUs per droplet.

*D. melanogaster* microbiota does not easily grow under the above culturing conditions (e.g. ref. [42]) Nonetheless we homogenised control flies (Ringer's injected and naïve) as a control. We rarely retrieved foreign CFUs after homogenising Ringer's injected or naïve flies (23 out of 642 cases, i.e., 3.6 %). We also rarely observed contamination in the bacteria-injected flies: except for homogenates from 27 out of 1223 flies (2.2 %), colony morphology and colour were always consistent with the injected bacteria (see methods of ref. [65]). Twenty one of these 27 flies were excluded from further analyses given that the contamination made counts of the injected bacteria unreliable; the remaining six flies had only one or two foreign CFUs in the most concentrated homogenate dilution, therefore these flies were included in further analyses. For *L. lactis* (70 out of 321 flies), *P. burhodo-granaeria* (7 out of 381 flies) and *Ps. entomophila* (1 out of 71 flies) there

were too many CFUs to count at the highest dilution. For these cases, we denoted the flies as having the highest countable number of CFUs found in any fly for that bacterium and at the highest dilution[23]. This will lead to an underestimate of the bacterial load in these flies. Note that because the assay is destructive, bacterial loads were measured once per fly.

## Bacterial load of dead flies

For two periods of time in the chronic infection phase, i.e., between 14 and 35 days and 56 to 78 days post injection, dead flies were retrieved from their vial at the daily survival checks and homogenised in order to test whether they died whilst being infected, or whether they had cleared the infection before death. The fly homogenate was produced in the same way as for live flies, but we increased the dilution of the homogenate (1:1 to 1:$10^{12}$) because we anticipated higher bacterial loads in the dead compared to the live flies. The higher dilution allowed us more easily to determine whether there was any obvious contamination from foreign CFUs or not. Because the flies may have died at any point in the 24 h preceding the survival check, and the bacteria can potentially continue replicating after host death, we evaluated the infection status (yes/no) of dead flies instead of the number of CFUs. Dead flies were evaluated for two experimental replicates per bacteria, and 160 flies across the whole experiment. Similar to homogenisation of live flies, we rarely observed contamination from foreign CFUs in the homogenate of dead bacteria-injected flies (3 out of 160; 1.9 %); of these three flies, one fly had only one foreign CFU, so it was included in the analyses. Dead Ringer's injected and naïve flies were also homogenised and plated as controls, with 6 out of 68 flies (8.8%) resulting in the growth of unidentified CFUs.

## Statistical analyses

Statistical analyses were performed with R version 4.2.1[66] in RStudio version 2022.2.3.492[67]. The following packages were used for visualising the data: "dplyr"[68], "ggpubr"[69], "gridExtra"[70], "ggplot2"[71], "plyr"[72], "purr"[73], "scales"[74], "survival"[75,76], "survminer"[77], "tidyr"[78] and "viridis"[79], as well as Microsoft PowerPoint for Mac v16.60 and Inkscape for Mac v 1.0.2. Residuals diagnostics of the statistical models were carried out using "DHARMa"[80], analysis of variance tables were produced using "car"[81], and post-hoc tests were carried out with "emmeans"[82]. To include a factor as a random factor in a model it has been suggested that there should be more than five to six random-effect levels per random effect[83], so that there are sufficient levels to base an estimate of the variance of the population of effects[84]. In our experimental designs, the low numbers of levels within the factors 'experimental replicate' (two to three levels) and 'person' (two levels), meant that we therefore fitted them as fixed, rather than random factors[84]. However, for the analysis of clearance (see below) we included species as a random effect because it was not possible to include it as a fixed effect because PPP is already a species-level predictor. Below we detail the statistical models that were run according to the questions posed. All statistical tests were two-sided.

**Do the bacterial species differ in virulence?** To test whether the bacterial species differed in virulence, we performed a linear model with the natural log of the maximum hazard as the dependent variable and bacterial species as a factor. Post-hoc multiple comparisons were performed using "emmeans"[82] and "magrittr"[85], using the default Tukey adjustment for multiple comparisons. Effect sizes given as Cohen's *d*, were also calculated using "emmeans", using the sigma value of 0.4342, as estimated by the package. The hazard function in survival analyses gives the instantaneous failure rate, and the maximum hazard gives the hazard at the point at which this rate is highest. We extracted maximum hazard values from time of death data for each bacterial species/dose/experimental replicate. Each maximum hazard

per species/dose/experimental replicate was estimated from an average of 33 flies (a few flies were lost whilst being moved between vials etc.). To extract maximum hazard values we defined a function that used the "muhaz" package[86] to generate a smooth hazard function and then output the maximum hazard in a defined time window, as well as the time at which this maximum is reached. To assess the appropriate amount of smoothing, we tested and visualised results for four values (1, 2, 3 and 5) of the smoothing parameter, $b$, which was specified using bw.grid[87]. We present the results from $b = 2$, but all of the other values gave qualitatively similar results (see Supplementary Table 2). We used bw.method = "global" to allow a constant smoothing parameter across all times. The defined time window was zero to 20 days post injection. We removed one replicate (92 CFU for *E. cloacae* infection) because there was no mortality in the first 20 days and therefore the maximum hazard could not be estimated. This gave final sizes of $n = 14$ for *E. cloacae* and $n = 15$ for each of the other three species.

$$\text{Model 1}: \log(\text{maximum hazard}) \sim \text{bacterial species}$$

**Are virulence differences due to variation in pathogen exploitation or PPP?** To test whether the bacterial species vary in PPP, we performed a linear model with the natural log of the maximum hazard as the dependent variable, bacterial species as a factor, and the natural log of infection intensity as a covariate. We also included the interaction between bacterial species and infection intensity: a significant interaction would indicate variation in the reaction norms, i.e., variation in PPP. The package "emmeans"[82] was used to test which of the reaction norms differed significantly from each other. We extracted maximum hazard values from time of death data for each bacterial species/dose/experimental replicate as described in section "Do the bacterial species differ in virulence?". We also calculated the maximum hazard for the Ringer's control groups, which gives the maximum hazard in the absence of infection (the y-intercept). We present the results from $b = 2$, but all of the other values gave qualitatively similar results (see results). We wanted to infer the causal effect of bacterial load upon host survival (and not the reverse), therefore we reasoned that the bacterial load measures should derive from flies homogenised before the maximum hazard had been reached. For *E. cloacae*, *L. lactis*, and *Pr. burhodogranariea*, for all smoothing parameter values, the maximum hazard was reached after two days post injection, although for smoothing parameter value 1, there were four incidences where it was reached between 1.8- and 2-days post injection. Per species/dose/experimental replicate we therefore calculated the geometric mean of infection intensity combined for days 1 and 2 post injection. In order to include flies with zero load, we added one to all load values before calculating the geometric mean. Geometric mean calculation was done using the R packages "dplyr"[68], "EnvStats"[88], "plyr"[72] and "psych"[89]. Each mean was calculated from the bacterial load of eight flies, except for four mean values for *E. cloacae*, which derived from four flies each.

For *Ps. entomophila* the maximum hazard was consistently reached at around day one post injection, meaning that bacterial sampling happened at around the time of the maximum hazard, and we therefore excluded this bacterial species from the analysis. We removed two replicates (Ringer's and 92 CFU for *E. cloacae* infection) because there was no mortality in the first 20 days and therefore the maximum hazard could not be estimated. One replicate was removed because the maximum hazard occurred before day 1 for all $b$ values (92,000 CFU for *E. cloacae*) and six replicates were removed because there were no bacterial load data available for day one (experimental replicate three of *L. lactis*). This gave final sample sizes of $n = 15$ for *E. cloacae* and $n = 12$ for *L. lactis*, and $n = 18$ for *Pr. burhodogranariea*.

$$\text{Model 2}: \log(\text{maximum hazard}) \sim \log(\text{geometric mean bacterial load}) \times \text{bacterial species}$$

To test whether there is variation in pathogen exploitation (infection intensity measured as bacterial load), we performed a linear model with the natural log of infection intensity as the dependent variable and bacterial species as a factor. Similar to the previous model, we used the geometric mean of infection intensity combined for days 1 and 2 post injection, for each bacterial species/dose/experimental replicate. The uninfected Ringer's replicates were not included in this model. Post-hoc multiple comparisons were performed using "emmeans", using the default Tukey adjustment for multiple comparisons. Effect sizes given as Cohen's $d$, were also calculated using "emmeans", using the sigma value of 2.327, as estimated by the package. *Ps. entomophila* was excluded for the reason given above. The sample sizes per bacterial species were: $n = 13$ for *E. cloacae*, $n = 10$ for *L. lactis* and $n = 15$ for *Pr. burhodogranariea*.

$$\text{Model 3}: \log(\text{geometric mean bacterial load}) \sim \text{bacterial species}$$

**Are persistent infection loads dose-dependent?** We tested whether initial injection dose is a predictor of bacterial load at seven days post injection[22,25]. We removed all flies that had a bacterial load that was below the detection limit as they are not informative for this analysis. The response variable was natural log transformed bacterial load at seven days post-injection and the covariate was natural log transformed injection dose, except for *P. burhodogranariea*, where the response variable and the covariate were log-log transformed. Separate models were carried out for each bacterial species. Experimental replicate and person were fitted as fixed factors. By day seven none of the flies injected with 92,000 CFU of *L. lactis* were alive. The analysis was not possible for *Ps. entomophila* infected flies because all flies were dead by seven days post injection.

$$\text{Model 4}: \log(\text{day 7 bacterial load}) \sim \log(\text{injection dose}) + \text{replicate} + \text{person}$$

**Calculation of clearance indices.** To facilitate the analyses of clearance we calculated clearance indices, which aggregate information about clearance into a single value for each bacterial species/dose/experimental replicate. All indices were based on the estimated proportion of cleared infections (defined as samples with a bacterial load that was below the detection limit) of the whole initial population. For this purpose, we first used data on bacterial load in living flies to calculate the daily proportion of cleared infections in live flies for the days that we sampled. Then we used the data on fly survival to calculate the daily proportion of flies that were still alive. By multiplying the daily proportion of cleared flies in living flies with the proportion of flies that were still alive, we obtained the proportion of cleared infections of the whole initial population – for each day on which bacterial load was measured. We then used these data to calculate two different clearance indices, which we used for different analyses. For each index we calculated the mean clearance across several days. Specifically, the first index was calculated across days three and four post injection (clearance index$_{3,4}$), and the second index was calculated from days seven, 14 and 21 (clearance index$_{7,14,21}$).

**Do the bacterial species differ in clearance?** To test whether the bacterial species differed in clearance, we used clearance index$_{3,4}$, which is the latest timeframe for which we could calculate this index for all four species: due to the high virulence of *Ps. entomophila* we were not able to assess bacterial load and thus clearance for later days. The distribution of clearance values did not conform to the assumptions of a linear model. We therefore used a Kruskal-Wallis test with pairwise Mann-Whitney-U post hoc tests. Note that the Kruskal-Wallis test uses a Chi-square distribution for approximating the H test statistic. To control for multiple testing we corrected the p-values of the post hoc tests using the method proposed by Benjamini and

Hochberg[90] that is implemented in the R function *pairwise.wilcox.test*.

$$\text{Model 5 : clearance index}_{3,4} \sim \text{bacterial species}$$

**Do exploitation or PPP predict variation in clearance?** To assess whether exploitation or PPP predict variation in clearance we performed separate analyses for clearance index$_{3,4}$ and clearance index$_{7,14,21}$. As discussed above, this precluded analysing *Ps. entomophila*. For each of the two indices we fitted a linear mixed effects model with the clearance index as the response variable. As fixed effects predictors we used the replicate-specific geometric mean log bacterial load and the species-specific PPP. In addition, we included species as a random effect.

In our analysis we faced the challenge that many measured clearance values were at, or very close to zero. In addition, clearance values below zero do not make conceptual sense. To appropriately account for this issue, we used a logit link function (with Gaussian errors) in our model, which restricts the predicted clearance values to an interval between zero and one. Initial inspections of residuals indicated violations of the model assumption of homogenously distributed errors. To account for this problem, we included the log bacterial load and PPP as predictors of the error variance, which means that we used a model in which we relaxed the standard assumption of homogenous errors and account for heterogenous errors by fitting a function of how errors vary. For this purpose, we used the option *dispformula* when fitting the models with the function *glmmTMB*[91].

$$\text{Model 6 : clearance index}_{3,4} \text{ or clearance index}_{7,14,21}$$
$$\sim \log(\text{geometric mean bacterial load}) + \text{PPP} + \text{bacterial species}_{random}$$

**Does longer-term clearance depend upon the injection dose?** In contrast to the analyses described above, we additionally aimed to assess the long-term dynamics of clearance based on the infection status of dead flies collected between 14 and 35 days and 56 to 78 days after injection. Using binomial logistic regressions, we tested whether initial injection dose affected the propensity for flies to clear an infection with *E. cloacae* or *Pr. burhodogranariea* before they died. The response variable was binary whereby 0 denoted that no CFUs grew from the homogenate and 1 denoted that CFUs did grow from the homogenate. Log-log transformed injection dose was included as a covariate as well as its interaction with the natural log of day post injection, and person was fitted as a fixed factor. Replicate was included in the *Pr. burhodogranariea* analysis only, because of unequal sampling across replicates for *E. cloacae*. *L. lactis* injected flies were not analysed because only 4 out of 39 (10.3%) cleared the infection. *Ps. entomophila* infected flies were not statistically analysed because of a low sample size ($n = 12$). The two bacterial species were analysed separately.

$$\text{Model 7 : CFU presence/absence}_{dead} \sim \log(\log(\text{injection dose}))$$
$$\times \log(\text{day post injection}) + \text{replicate} + \text{person}$$

To test whether the patterns of clearance were similar for live and dead flies we tested whether the proportion of live uninfected flies was a predictor of the proportion of dead uninfected flies. We separately summed up the numbers of uninfected and infected flies for each bacterial species and dose, giving us a total sample size of $n = 20$ (four species × five doses). For live and for dead homogenised flies we had a two-vector (proportion infected and proportion uninfected) response variable, which was bound into a single object using cbind. The predictor was live flies, and the response variable was dead flies, and it was analysed using a generalized linear model with family = quasibinomial.

$$\text{Model 8 : cbind(dead uninfected, dead infected)} \sim \text{cbind(live uninfected, live infected)}$$

**Reporting summary**
Further information on research design is available in the Nature Research Reporting Summary linked to this article.

## Data availability
The minimum datasets generated in this study are available on Refubium, the institutional repository of the Freie Universität Berlin (https://doi.org/10.17169/refubium-35174[92]). Along with the below code, the datasets can be used to produce all statistical results and the data figures.

## Code availability
The R code used to process the datasets and produce the data figures for this study is available on Refubium, the institutional repository of the Freie Universität Berlin (https://doi.org/10.17169/refubium-35174[92]).

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

## Acknowledgements
We thank the Hiesinger group, the Rolff group, Karsten Karczewski, Luisa Linke, Alexandro Rodríguez-Rojas, Jens Rolff and Seulkee Yang for advice and/or technical support. We thank Jens Rolff for feedback on earlier drafts of the manuscript. This work was supported by the grants AR 872/3-1, AR 872/4-1, AR 872/5-1 (Research Unit FOR 5026) and AR 872/7-1 awarded to SAOA by the Deutsche Forschungsgemeinschaft (DFG; https://www.dfg.de/en/index.jsp). MF and RRR were supported by the DFG Research Unit FOR 5026, spokesperson/grant awarded to Jens Rolff.

## Author contributions
S.A.O.A., conceived the overall idea. B.A.H., L.S. and S.A.O.A. designed the experiments and collected the data. B.A.H., L.S., M.F. and S.A.O.A. wrote the manuscript. M.F. & R.R.R. conceived the virulence decomposition and clearance analyses and M.F., R.R.R. & S.A.O.A. analysed the data. All authors contributed critically to the drafts.

## Funding

## Competing interests
The authors declare no competing interests.
