## [Peer Review File · Nature Communications]

Reviewer comments, initial review

Reviewer #1 (Remarks to the Author):

In this paper, the authors set up the idea that clearance/resistance versus chronicity/persistence depends on the costs and benefits of immunity, and the benefits of immunity depend on both parasite exploitation and per-parasite pathogenicity. They set out to test that prediction using several different pathogenic microbes of *D. melanogaster*. I found the conceptual framework intuitive and compelling and the experiments and data insightful and interesting.

The fundamental problem that I have with the paper is that I do not think that the conceptual framework set out in the Introduction can be tested with the data gathered here. This is for two related reasons. First, it suggests that the only (or at least the most likely) outcomes of infection are either a persistent infection or clearance. Second, it suggests that the costs of infection can be less than the costs of immunity, so that, in the face of trade-offs between immunity and other life history traits, it makes sense for the host to not invest in immunity. But in this dataset, host death is a more frequently observed outcome than persistent infections, and it is not clear to me that this framework makes any sense for thinking about infections that can be fatal. So, for example, on lines 235-237, the authors say, "as the mean bacterial load increased, clearance later in the infection decreased, which supports our second prediction," referring to the prediction that, "if costs of clearance increase faster than benefits, then we predicted that increasing exploitation results in decreasing clearance." To me, it doesn't make sense to talk about the costs of clearance increasing faster than the benefits of clearance, when the cost of failing to clear the infection is death. I suppose the argument I'm making is somewhat semantic – if a host is unable to mount a strong enough immune response to clear an infection and ends up dying, perhaps it is appropriate to say that it was unable to pay the cost of clearance. But I think the way these costs and benefits are discussed in the Introduction is set up to think about the costs and benefits of clearance in a life-history theory framework, and death doesn't really fit into that picture.

The other challenge with this conceptual framework is that there is no data on immunity, which seems critical to understanding the supposed costs and benefits of clearance. For example, how would your interpretation of the results differ from what you have presented if you found that the immune responses were actually weaker in individuals/treatments/parasites with high bacterial load and low clearance than they were in cases where clearance occurred? What if that pattern was reversed? It's actually not even clear to me which pattern would provide stronger support for your conclusions.

This data may present an interesting opportunity to propose a different framework for thinking about the three possible outcomes of infection: a persistent, non-fatal infection; clearance; and a fatal infection. I am unaware of any conceptual framework that can tease apart the contributions of immunity, pathogen growth, and per-parasite pathogenicity to each of these three outcomes. I would encourage the authors to think more about that problem.

Minor comments:

Lines 38-45: You lead with by introducing resistance as the ability of the host to limit pathogen within-host growth, but then shift to pathogen persistence. I think there is a potential problem here, in that persistence and resistance are not necessarily identical; that is, a pathogen can persist, even if its growth is limited. You could deal with this by focusing the definition of resistance onto the ability of the immune response to clear an infection, rather than just limit growth.

Line 41: Unnecessary comma after "traits"

Line 42: Unnecessary comma after "persists"

Line 47-49: This line seems disconnected from the rest of the paragraph. Moreover, this is not the only reason that persistent infections might matter, and since your work doesn't actually deal with transmission, saying that this is the only reason that persistence matters seems like it undercuts your work.

Lines 47-53: This paragraph contains a fair bit of insect- and *Drosophila*-specific detail that somewhat undercuts your main argument. In particular, this paragraph makes it seem like you

don't really know how to define a chronic infection in *Drosophila*.

Lines 56-57: "the factors that cause persistence or clearance are not well-understood" Is this meant to be a general statement, or a *Drosophila*- or insect-specific statement? For example, the drivers of persistence versus chronicity are fairly clear for vertebrate-helminth infections.

Line 59: Virulence has many definitions, but saying that it is "disease severity" is not helpful, since "severity" is just as vague as "virulence." Just say that you are defining virulence as the decrease in fitness caused by infection.

Line 63: "side effect" sounds like an accidental by-product; you mean that virulence is caused by increased exploitation due to higher load

Lines 85-87: This is a key sentence, but I found it hard to follow as currently written.

Lines 91-99: Where and when were the pathogens isolated? How long has each been in culture? What is the protocol for maintaining the pathogens in the lab? I am asking because I wonder how long each has had to adapt to infecting flies in a lab, and whether than adaptation might have involved unintentional selection for higher virulence.

Fig. 3: One thing that appears evident in these plots is that the pathogen growth rate is quite different across pathogens and infection doses. For example, for *Providentia* infections, the 920 and 1840 dose treatments showed considerable growth by Day 1, whereas the 92000 dose treatment appeared not to grow at all. I wonder whether a better measure (or at least different) measure of exploitation would be pathogen growth rate. Could a lack of growth in the highest doses suggest that exploitation is actually lower there than at a lower dose?

Lines 160-165: It's unclear where the flies for this analysis came from.

Fig. 5: In cases where you find individuals that cleared the infection at one age, and then there are zero flies available at later ages (such as in panel D), does that suggest/indicate that, while there were some flies that had cleared the infection by age 2, no individuals were alive by age 4? That might suggest that some of the dead individuals at age 4 had actually cleared the infection, whereas some had not. Could that be used as evidence of an extreme cost of clearance? E.g., there were flies that cleared the infection, but the cost of that clearance was so high that the flies died?

Reviewer #2 (Remarks to the Author):

The ms "Decomposing virulence to understand bacterial clearance in persistent infections" present the results of a series of bacterial infections in the fruit fly *Drosophila melanogaster*, using 4 different bacterial species of different virulence, at different doses. From these experimental infections, authors measured bacterial exploitation (from bacterial load), per pathogen pathogenicity (PPP) from the relationship between virulence (measured from host mortality) and bacterial load, persistence and clearance of the bacteria. These measures enabled to define the specific host-pathogen interactions for each bacterial species, and shed new light on how we may define the different types and evolutionary strategies of pathogens.

I found the paper well written, very pleasant to read, and the experimental work smart and elegant. The choice of bacterial species, including a low virulence bacteria naturally present as a member of gut microbiome in *Drosophila*, was clever, and served the purpose of the work nicely. I appreciate how authors introduced the concepts, including with figure 1, which greatly facilitated the understanding of the large amount of data analysis that followed. I also enjoyed how authors provide novel data to demonstrate the cost of clearance when bacterial load increases.

I do not have any negative comments about this manuscript. I just want to mention that, if I clearly praise authors for this work, and the significant contribution in the understanding of host-pathogen interactions, I may have been a little frustrated not to find in the discussion a little more about how this concept can be broadly applicable, as mentioned in the abstract. For example, what could be the implication of such framework in better understanding specific diseases, or pathogenic strains, in medicine or veterinary sciences?

But overall, I really enjoyed the read, and I am sure this paper will be well received in the field.

Below are my minor comments:

- Figure 3: I was wondering why bacterial loads for *L. lactis* went to the roof? Did authors encounter a problem of quantification for high loads during the assay? What is the impact on further analyses? Also, line 175, authors mentioned a possible underestimation of bacterial load.

Again, how did this impact the results in the paper?

- I am a bit skeptical about the use of 5 infected flies in the same vial to unambiguously measure bacterial persistence. For some bacteria, the clearance level was high, and in such condition, I can imagine that some flies may have cleared the disease and went re-infected from other individuals. Can authors comment on this please?

- Figure 6A: What statistical analysis was performed for the comparison of clearance index across bacterial species? In the text, I can see a chi-square score, whilst in the figure caption, a non-parametric test has been applied. Please clarify.

RESPONSE TO REVIEWERS' COMMENTS (Acuña Hidalgo et al)

The authors' numbered replies are given in blue text below the original comments from the reviewers.

Reviewer #1

In this paper, the authors set up the idea that clearance/resistance versus chronicity/persistence depends on the costs and benefits of immunity, and the benefits of immunity depend on both parasite exploitation and per-parasite pathogenicity. They set out to test that prediction using several different pathogenic microbes of *D. melanogaster*. I found the conceptual framework intuitive and compelling and the experiments and data insightful and interesting.

1. Thank you for your positive feedback and your very helpful suggestions to improve the manuscript.

The fundamental problem that I have with the paper is that I do not think that the conceptual framework set out in the Introduction can be tested with the data gathered here. This is for two related reasons. First, it suggests that the only (or at least the most likely) outcomes of infection are either a persistent infection or clearance. Second, it suggests that the costs of infection can be less than the costs of immunity, so that, in the face of trade-offs between immunity and other life history traits, it makes sense for the host to not invest in immunity. But in this dataset, host death is a more frequently observed outcome than persistent infections, and it is not clear to me that this framework makes any sense for thinking about infections that can be fatal. So, for example, on lines 235-237, the authors say, "as the mean bacterial load increased, clearance later in the infection decreased, which supports our second prediction," referring to the prediction that, "if costs of clearance increase faster than benefits, then we predicted that increasing exploitation results in decreasing clearance." To me, it doesn't make sense to talk about the costs of clearance increasing faster than the benefits of clearance, when the cost of failing to clear the infection is death. I suppose the argument I'm making is somewhat semantic – if a host is unable to mount a strong enough immune response to clear an infection and ends up dying, perhaps it is appropriate to say that it was unable to pay the cost of clearance. But I think the way these costs and benefits are discussed in the Introduction is set up to think about the costs and benefits of clearance in a life-history theory framework, and death doesn't really fit into that picture.

2. The reviewer raises a number of related concerns regarding our conceptual framework. It is our understanding that these concerns have arisen because of some imprecise and undetailed descriptions in our original framework.

In our revised manuscript, we have now explicitly distinguished between (1) host clearance effort, which relates to the process of (attempted) clearance, and (2) clearance rate, which relates to the successful outcome of the clearance process. In addition to rewording the text in the introduction and the discussion, we have constructed two new figures (fig. 2 & supplementary fig. 1) as well as a supplement to the fig. 2 legend in the supporting information (Supplementary legend for fig. 2) to facilitate understanding of our conceptual framework and the related derivation of our predictions.

To explain in more detail: The comments of the reviewer made us realise that in the previous manuscript version we had not sufficiently distinguished between the process and the outcome of clearance. As highlighted by the reviewer, we talked about the "costs of clearance", by which we meant the costs related to the effort to clear the pathogen. This does not guarantee pathogen clearance, and through, e.g., its potential effect on immunopathology it might even lead to host death. However, we suspect that the reviewer might have understood it to mean costs related to having achieved clearance. In our

understanding, a more outcome-focussed view could explain the concerns of the reviewer, most notably the issue of how death fits in.

The revised text in the introduction can be mostly found on pages 5-7 (lines 88-163), and in the discussion on pages 21-22 (lines 404-435).

The other challenge with this conceptual framework is that there is no data on immunity, which seems critical to understanding the supposed costs and benefits of clearance. For example, how would your interpretation of the results differ from what you have presented if you found that the immune responses were actually weaker in individuals/treatments/parasites with high bacterial load and low clearance than they were in cases where clearance occurred? What if that pattern was reversed? It's actually not even clear to me which pattern would provide stronger support for your conclusions.

3. In our manuscript we present a novel conceptual framework (see response number 4). We can indeed use this framework to formulate some basic predictions about immunity, which is something that we had not done in the previous version of our manuscript. These predictions can be derived from the newly added figure 2. In the previous version of our manuscript, we predicted that PPP should result in an increased clearance rate. We propose that this effect is mechanistically mediated by an increase in clearance effort of the host (fig. 2f & i, Hypothesis 1). Accordingly, we could now predict that such an increased host clearance effort should be apparent in terms of a more intense immune response. The same argumentation applies to our first prediction for exploitation (fig. 2f & i, Hypothesis 2), which we argued should also result in an increased clearance rate. Note that our understanding of immunity in this context is the realised immune response and its potential costs, and not the different components of the immune system. In contrast, our second prediction for exploitation (fig. 2f & n, Hypothesis 3) is based on the idea that increased exploitation makes it harder to clear the pathogen and thus results in a decreased clearance rate. Importantly, this effect is not mediated by a change in host clearance effort – and thus should not result in a change in immunity.

However, we would like to point out that our framework (necessarily) does not attempt to take the full complexities of host-pathogen interactions into account. As examples to explain what we mean by complexities, two experimental/biological issues would be (1) At what point during the infection process one should assay the immune response? And (2) Which aspects of the immune response (e.g., cellular and/or humoral response) should be assayed? Bacterial infections are dynamic processes, where bacterial load (e.g., Duneau et al 2017 DOI: <https://doi.org/10.7554/eLife.28298>) and the host immune response towards the pathogen can change over time, and where they vary according to the infecting microorganism (as seen in immune gene expression data; e.g., Lemaitre et al 1997 PNAS 94, 14614). These kinds of complexities are intentionally not included in our model, because the measured parameters will likely vary depending upon the host-pathogen interaction being considered. Therefore, we suggest that one would need to make predictions based upon knowledge about the immune responses towards the pathogen that is being studied, and how these fluctuate over time.

Regarding including additional data on immunity: Whilst we agree that this would be interesting, we feel that such an effort is a new research endeavour on its own and is beyond the scope of this manuscript, which is to understand how pathogen-related factors and host clearance effort interact to affect virulence. Nevertheless, we think that in order to stimulate future research on patterns of immunity, it would be important to include the predictions we have outlined in the paragraph above, and also to suggest the kinds of complexities that would need to be considered. We have added the following text to the discussion on page 22 (lines 437-448):

“The conceptual framework that we present here can be used to formulate some basic predictions about immunity. For example, we proposed that an increased clearance rate is mechanistically mediated by an increased host clearance effort (H1 and H2), and we could predict that this increased clearance effort would be due to a stronger immune response. In contrast, H3 is not mediated by a change in host clearance effort – and thus is not predicted to result in a change in immune response. We also note that these hypotheses are not necessarily mutually exclusive. However, the framework (necessarily) does not attempt to take the full complexities of host-pathogen interactions into account. For example, bacterial infections are dynamic processes, where bacterial load [e.g., 25] and the host immune response towards the pathogen can change over time, and where they vary according to the infecting microorganism [60]. These parameters will also likely vary depending upon the host-pathogen interaction being considered.”

This data may present an interesting opportunity to propose a different framework for thinking about the three possible outcomes of infection: a persistent, non-fatal infection; clearance; and a fatal infection. I am unaware of any conceptual framework that can tease apart the contributions of immunity, pathogen growth, and per-parasite pathogenicity to each of these three outcomes. I would encourage the authors to think more about that problem.

4. Our apologies – we had not clearly or explicitly detailed the theoretical/conceptual framework in the previous version of our manuscript. We agree that it would be an excellent opportunity, and we have now done so.

In detail, the manuscript includes the three outcomes of infection (fatal infection, persistent infection and cleared infection) and proposes how pathogen traits (PPP and exploitation) and a host trait (immunity-based host clearance effort) affect the likelihood of each outcome. In our response number 2 above, we have expanded on our explanation of the framework and its implications, and we have hopefully clarified some confusing issues. In this context the new figure (fig. 2f) illustrates our conceptual view on how pathogen and host traits affect virulence and clearance rate. In addition, this figure includes links to the three different infection outcomes mentioned above.

We have modified the text in the introduction as explained in response number 2, and we have mentioned in the final paragraph of the discussion that this novel framework is a conceptual contribution to the field (page 22, lines 453-455) (also in response to reviewer 2, reply number 19).

Minor comments:

Lines 38-45: You lead with by introducing resistance as the ability of the host to limit pathogen within-host growth, but then shift to pathogen persistence. I think there is a potential problem here, in that persistence and resistance are not necessarily identical; that is, a pathogen can persist, even if its growth is limited. You could deal with this by focusing the definition of resistance onto the ability of the immune response to clear an infection, rather than just limit growth.

5. In the previous version of our manuscript we had not explicitly defined the link between resistance and clearance, and we think that doing so will help the transition from resistance to persistence. Given that resistance can also be defined as a reduction in infection load and not only clearance, we have therefore opted to include the following sentence in the second sentence of the introduction at lines 40-41: “Reduced pathogen growth can be beneficial to the host by limiting pathogen load and/or facilitating pathogen clearance”.

Line 41: Unnecessary comma after “traits”

6. The comma has been removed.

Line 42: Unnecessary comma after “persists”

7. The comma has been removed.

Line 47-49: This line seems disconnected from the rest of the paragraph. Moreover, this is not the only reason that persistent infections might matter, and since your work doesn't actually deal with transmission, saying that this is the only reason that persistence matters seems like it undercuts your work.

8. This is a good point. We have adjusted the sentence so that we give some more examples as to why persistence is relevant, and we have also rephrased the last sentence to improve the connectedness of this paragraph. The corrections can be found on page 3, lines 49-54:

“Persistent infections can have important consequences, e.g., prolonged host effort to clear the infection, accumulation of infection costs for the host, increased chance for within-host pathogen evolution, and increased pathogen transmission in the host population. These phenomena could also have wide-reaching effects given the broad range of taxa found to sustain persistent bacterial infections, e.g., humans [12], other vertebrates [13-16], and insects [17].”

Lines 47-53: This paragraph contains a fair bit of insect- and *Drosophila*-specific detail that somewhat undercuts your main argument. In particular, this paragraph makes it seem like you don't really know how to define a chronic infection in *Drosophila*.

9. To make the taxonomic representation a little more balanced, we have included some examples of persistent infections in vertebrate taxa other than humans (line 53).

Furthermore, we have rephrased the last sentence about *Drosophila*. Our intention in that sentence was not to define a chronic infection as lasting seven days. Rather, seven days was meant as the lower cut-off of the duration of experiments that had been performed to study this. The rephrased sentence reads as follows and is found on page 3, lines 55-57:

“Disparate bacterial species have been shown to persist for at least seven days inside the host species used in this study, *D. melanogaster* [19, 20, 22-27]”

Lines 56-57: “the factors that cause persistence or clearance are not well-understood” Is this meant to be a general statement, or a *Drosophila*- or insect-specific statement? For example, the drivers of persistence versus chronicity are fairly clear for vertebrate-helminth infections.

10. Our apologies - we were vague in our previous formulation. On page 3 (lines 59-61) we have rephrased this to: “we have a limited understanding of the relationship between the different infection outcomes, i.e., fatal infection, persistent infection or clearance of infection, especially in insects”.

Line 59: Virulence has many definitions, but saying that it is “disease severity” is not helpful, since “severity” is just as vague as “virulence.” Just say that you are defining virulence as the decrease in fitness caused by infection.

11. We have rephrased this sentence as suggested. See the beginning of page 4, lines 63-64.

Line 63: “side effect” sounds like an accidental by-product; you mean that virulence is caused by increased exploitation due to higher load

12. We have removed “side effect” from this sentence and slightly rephrased this part of the sentence (see page 4, line 67) to: “where an increase in virulence is caused by an increase in pathogen load”

Lines 85-87: This is a key sentence, but I found it hard to follow as currently written.

13. We have removed this sentence and instead rewritten parts of the introduction to explain our conceptual framework and the associated hypotheses and predictions in more detail. Please also see our responses numbers 2 and 4.

Lines 91-99: Where and when were the pathogens isolated? How long has each been in culture? What is the protocol for maintaining the pathogens in the lab? I am asking because I wonder how long each has had to adapt to infecting flies in a lab, and whether than adaptation might have involved unintentional selection for higher virulence.

14. This is an interesting point. A host is not required for the maintenance of any of the species of bacteria that we used, meaning that during maintenance the bacteria were not subjected to host selection which may have resulted in increased virulence. We store cultures of the original stocks, which were provided to us by other labs or the German collection of microorganisms, at -80 °C. We have added a sentence to the materials and methods on page 23 (lines 481-483) that reads: “All bacterial species were stored in 34.4% glycerol at -80 °C and new cultures were grown freshly for each experimental replicate.” For more details about where and when the strains were isolated please see the below table:

Bacterial species	Where isolated?	When isolated?	References (all the below references are given in the materials and methods)
E. cloacae	Maize plant, unknown country of origin	Before 1980	https://www.dsmz.de/collection/catalogue/details/culture/DSM-16657
L. lactis	D. melanogaster captured in an apple orchard (Pennsylvania, US)	1998 and 2001	Lazzaro 2002
Pr. burhodogranariea	D. melanogaster captured in an apple orchard (Pennsylvania, US)	2008	Juneja & Lazzaro 2009
Ps. entomophila	D. melanogaster captured in Guadeloupe	2005	Vodovar et al 2005

Fig. 3: One thing that appears evident in these plots is that the pathogen growth rate is quite different across pathogens and infection doses. For example, for *Providentia* infections, the 920 and 1840 dose treatments showed considerable growth by Day 1, whereas the 92000 dose treatment appeared not to grow at all. I wonder whether a better measure (or at least different) measure of exploitation would be pathogen growth rate. Could a lack of growth in the highest doses suggest that exploitation is actually lower there than at a lower dose?

15. Due to the insertion of an additional figure (Fig. 2), Fig. 3 is now Fig. 4 in the revised manuscript.

We agree with the reviewer that there is quite some variation across pathogens and infection doses in the bacterial load compared to the original infection dose. However, from our point of view, our data is not suitable for identifying variation in growth rates. Our first sampling point is 24-hours post infection, and we cannot discount the possibility that the peak bacterial load was reached earlier than this, so any patterns that suggest lower bacterial loads than the initial injection dose, could be explained by the fact that the bacterial numbers are already declining. However, we cannot know this because we did not measure pathogen load before 24 hours. Moreover, the variation between flies injected with each bacteria/dose is often large. The pattern mentioned by the reviewer is mainly apparent when looking at the medians but it seems to largely disappear when considering the distributions of all data points. In the figure we had previously plotted medians, which contrasts with the use of geometric means in our subsequent analyses. To be more consistent we have now changed this, and plotted geometric means in the figure. After this change the pattern identified by the reviewer also became much less apparent. For these reasons we currently see no clear support for the pattern identified by the reviewer.

Furthermore, the definition of exploitation that we have used in this manuscript follows the description of the term by Råberg and Stjernman (2012), i.e., it affects infection intensity, which we have measured as bacterial load. Therefore, to keep with the original framework of the virulence decomposition we would like to keep using this definition of the response variable rather than using pathogen growth rate.

Lines 160-165: It's unclear where the flies for this analysis came from.

16. We have inserted an extra sentence starting on page 14 (lines 250-251) to explain where the flies originated from: "These dead flies originated from daily survival checks made during the experiment to estimate virulence". More details can be found in the materials and methods in the section (lines 576-579).

Fig. 5: In cases where you find individuals that cleared the infection at one age, and then there are zero flies available at later ages (such as in panel D), does that suggest/indicate that, while there were some flies that had cleared the infection by age 2, no individuals were alive by age 4? That might suggest that some of the dead individuals at age 4 had actually cleared the infection, whereas some had not. Could that be used as evidence of an extreme cost of clearance? E.g., there were flies that cleared the infection, but the cost of that clearance was so high that the flies died?

17. The data were collected cross-sectionally by sacrificing living flies from vials of flies that were separate from the vials of flies used for survival. If there are zero flies available for bacterial load at later ages, it is because we had already sampled all of the living flies, so it does not allow us to make any inferences about survival dynamics. We've added a sentence to the end of the figure 6 legend (page 16, lines 278-279) to hopefully clarify this point: "Zeros on the x-axis mean that there were no flies alive from which to assess clearance". From this data set we cannot say anything about what happened in the dead flies early in the infection. Although it is indeed possible that some dead flies (in the survival curves given in fig. 3) at 4-days post injection had cleared the infection, and that the death might indeed be due to an extreme cost of clearance. However we think that it is a little too speculative to include in the text.

Reviewer #2

The ms "Decomposing virulence to understand bacterial clearance in persistent infections" present the results of a series of bacterial infections in the fruit fly *Drosophila melanogaster*, using 4 different bacterial species of different virulence, at different doses. From these experimental infections, authors measured bacterial exploitation (from bacterial load), per pathogen pathogenicity (PPP) from the relationship between virulence (measured from host mortality) and bacterial load, persistence and clearance of the bacteria. These measures enabled to define the specific host-pathogen interactions for each bacterial species, and shed new light on how we may define the different types and evolutionary strategies of pathogens.

I found the paper well written, very pleasant to read, and the experimental work smart and elegant. The choice of bacterial species, including a low virulence bacteria naturally present as a member of gut microbiome in *Drosophila*, was clever, and served the purpose of the work nicely. I appreciate how authors introduced the concepts, including with figure 1, which greatly facilitated the understanding of the large amount of data analysis that followed. I also enjoyed how authors provide novel data to demonstrate the cost of clearance when bacterial load increases.

18. Thank you for your positive feedback and your very helpful suggestions to improve the manuscript.

I do not have any negative comments about this manuscript. I just want to mention that, if I clearly praise authors for this work, and the significant contribution in the understanding of host-pathogen interactions, I may have been a little frustrated not to find in the discussion a little more about how this concept can be broadly applicable, as mentioned in the abstract. For example, what could be the implication of such framework in better understanding specific diseases, or pathogenic strains, in medicine or veterinary sciences?

19. Thank you for pointing this out. We have now added more text to the concluding paragraph. The rephrased paragraph (page 22, lines 450-460) reads:

"In conclusion, our results strongly support that PPP is an important component driving variation in virulence, and that disentangling its contribution towards virulence, in combination with the contribution of exploitation, will undoubtedly help our mechanistic and evolutionary understanding of host-pathogen interactions. Our framework provides a general conceptual contribution to the field and as such could be applicable across a broad range of host-pathogen interactions. We suggest that such a decomposition of virulence can be used to better understand how virulence relates to other infection processes such as infection clearance and the effort that goes into clearance, and it may also contribute towards expanding the theory on virulence evolution. Furthermore when designing drugs in the medical and veterinary sciences, a better understanding of the contributions of PPP and exploitation towards virulence, may help to predict the evolutionary implications of a treatment."

But overall, I really enjoyed the read, and I am sure this paper will be well received in the field. Below are my minor comments:

- Figure 3: I was wondering why bacterial loads for *L. lactis* went to the roof? Did authors encounter a problem of quantification for high loads during the assay? What is the impact on further analyses? Also, line 175, authors mentioned a possible underestimation of bacterial load. Again, how did this impact the results in the paper?

20. For some flies, particularly those infected with *L. lactis* we unfortunately reached our (technical) upper detection limit. As mentioned in the materials and methods, it affected *L. lactis* (70 out of 321 flies) where all injection doses were affected and the effect was predominantly at day two post injection. It also affected *Pr. burhodogranaeria* (7 out of 381 flies) and *Ps. entomophila* (1 out of 71 flies). This meant that for these flies we had too many colony forming units to be able to reliably count, even at the highest dilution of the fly homogenate.

Given the low number of flies affected after injection with *P. burhodogranariea* and *Ps. entomophila*, the issue is unlikely to affect these analyses.

Although it is possible that it could have affected the *L. lactis* analyses, we argue that it is unlikely to have affected the interpretations of our data:

- i. Figure 5a, Parasite exploitation – Here an even higher bacterial load would be apparent for *L. lactis* than the one already shown. This would not change our interpretation of the results, but would rather act to exacerbate the difference in exploitation between *L. lactis* and the two other species of bacteria.
- ii. Figure 5b, PPP – Here a higher bacterial load would tend to drag the light pink data points further to the right, so it is unlikely to cause a pronounced change of the slope or our interpretation of the results.
- iii. Figure 7b and 7c – Here a higher load would again tend to drag the pink data points further to the right. Similarly to ii, this is unlikely to change our interpretation of the data.
- iv. Supp. Figure 4 – Two flies had too many CFUs to count in this analysis. Without knowing the exact bacterial load for these two flies, it is not possible to say whether their actual load would strengthen or weaken the relationship. However, we are already relatively cautious with this analysis because we did it both with and without these two flies (as explained in the main text at lines 265-267).

Regarding figures 5a and 5b we have added a sentence at lines 232-236 to say “The underestimate of the load of some of the *L. lactis*-injected flies (see methods) is unlikely to have affected the interpretations of the data in Fig. 5a: a higher load would have acted to exacerbate the higher load for *L. lactis*. For Fig. 5b a higher load would have dragged the pink data points further to the right so it may have flattened the slope.”

Regarding figures 7b and 7c we have added a sentence starting at line (page 17, lines 316-319) “The underestimate of the load of some of the *L. lactis*-injected flies is unlikely to have affected the interpretations of Figs. 7b and c: a higher load would have dragged the pink data points further to the right, giving a longer tail to the distribution.”

- I am a but skeptical about the use of 5 infected flies in the same vial to unambiguously measure bacterial persistence. For some bacteria, the clearance level was high, and in such condition, I can imagine that some flies may have cleared the disease and went re-infected from other individuals. Can authors comment on this please?

21. This is a good point. However, we think that direct transmission from dead to living flies is rather unlikely for the following reasons: We injected bacteria into the body cavity of the experimental flies. In order for transmission to occur from a dead to a living fly, the dead fly would first need to decompose in the food. In the absence of larvae in the food vials, we observed that the dead flies remain intact and do not appear to decompose in the time for which they are in the vial with living flies. We think that it is therefore unlikely that the living flies can contract an infection in this case. If, on the other hand, there are many larvae present in the food vials (fertilised eggs are produced during the first one to three-weeks post mating, i.e., up to the first three weeks after injection in these experiments) the food can be mixed up a bit by the larvae moving around inside it, and depending upon the number of larvae, after a few days the dead flies can start to move into the food during this process.

However, the experimental flies were moved into new vials before this stage was reached. Even if a fly had decomposed, it is rather unlikely that the bacteria will grow on the food, because the food contains a combination of two different antimicrobials: nipagin and propionic acid. We would also like to mention that keeping flies in groups after infection is a standard method in infections with *D. melanogaster* examining survival and bacterial load (e.g., Louie, Song, Hotson, Tate & Schneider 2016 [doi:10.1371/journal.pbio.1002435]; Chambers, Jacobson, Khalil, Lazzaro 2019 [https://doi.org/10.1371/journal.pone.0224440]; Iatsenko, Marra, Boquete, Peña, Lemaitre 2020 [https://doi.org/10.1073/pnas.1914830117]), indeed keeping flies in individual vials is uncommon, particularly when working with large sample sizes. We have included the above-mentioned references and the following text at lines 531-532: “Maintaining flies in groups after infection is a standard method in experiments with *D. melanogaster* that examine survival and bacterial load [e.g., 22, 63, 64].”

- Figure 6A: What statistical analysis was performed for the comparison of clearance index across bacterial species? In the text, I can see a chi-square score, whilst in the figure caption, a non-parametric test has been applied. Please clarify.

22. The figures have been renumbered, so that this comment now relates to figure 7a. We indeed used a Kruskal-Wallis test with pairwise Mann-Whitney-U post hoc tests, and then controlled for multiple testing using the method proposed by Benjamini and Hochberg. These details are given in the statistics section of the supplementary information, in paragraph 2 on page 14. There is a Chi-square value because the R function we used to perform the Kruskal-Wallis test uses a Chi-square distribution for approximating the H test statistic (most likely this approximation is done for computational efficiency as the calculation of the H test statistic can be computationally very demanding). We have added the following to the above-mentioned section of the supplementary information: “Note that the Kruskal-Wallis test uses a Chi-square distribution for approximating the H test statistic”.

Reviewer comments, second round review –

Reviewer #2 (Remarks to the Author):

I found the revised manuscript very clear and the response to every points raised by reviewers convincing. In particular, I think the broader description of the conceptual framework, with the inclusion of hypotheses H1 to H3, and their predictions P, as well as the concepts of clearance effort and clearance rate to account for significantly improved the paper. Although I found the figure 2 complex and a little overwhelming, I think the end of the introduction pages 7-8 explains it well.

This new figure 2 and the introduction of new clearance concepts serves the interpretation of the results and their discussion. However, I think more details could be given about why data support H3 and not H2. Sometimes I was left unsure how the data could actually support H3, and I would have appreciated a couple of sentences around Lines 312-315 to help me navigate the results and tell me why it does not support H2.

Minor comment:

The decision to represent a linear relationship between clearance rate and clearance effort in figure 1a, but an exponential relationship between immunity costs and clearance effort in 1c is not explicitly justified in the text. If I understand and mostly agree with this, I would have expected a line and/or a reference to back this up.

Reviewer #3 (Remarks to the Author):

In this manuscript Acuña Hidalgo et al present work aiming to partition the variance in virulence between host clearance and pathogen persistence. They employ bacterial infections in the fruit fly *Drosophila melanogaster*, using different doses of 4 bacterial species and measured bacterial exploitation (from bacterial load), and per pathogen pathogenicity (PPP) from the relationship between virulence (measured from host mortality) and bacterial load, persistence and clearance of the bacteria.

They use this data to define and establish a unifying framework for infection outcomes, where pathogen clearance or pathogen persistence are the outcomes of a balance between the relative strengths of host clearance efforts, the per-pathogen pathogenicity, and the pathogen level of host exploitation. To me, this unifying framework is the major strength of this work and an immensely valuable contribution to the field of infection and immunity. Hopefully, the days of reading papers that start with "host can either resist or tolerate infection" are soon over, and this framework explaining that both are possible and necessary outcomes will take hold in the field.

The paper overall is very well-written, and I enjoyed the clever and creative experimental work, including the choice of bacterial pathogens that (intentionally) span a range of levels of virulence. I also appreciated the large amount of data, making the paper very complete, including data on the cost of resistance/clearance, which I do not believe has been demonstrated in this way previously.

I have the benefit of reviewing a version of the manuscript that has already been reviewed and revised, and so unsurprisingly, my criticism, if any is minor. I did read the authors' responses to previous queries and was pleased to see how well they were able to address them all.

My only minor criticism is the lack of any data on immune gene expression. To be clear, this should not be a requirement for publication of this paper, which makes a fantastic contribution to the field with the framework. Nor does it change the way the framework is interpreted. Simply it would have added useful mechanistic insight into the mechanisms of clearance in operation during the acute and persistent phases of infection. Though this may be something the authors would prefer to explore separately, in much greater detail, instead of being buried and lost in this paper, which is already so data-rich.

Overall, I loved the paper, congratulations.

RESPONSE TO REVIEWERS' COMMENTS (Acuña Hidalgo et al)

The authors' numbered replies are given in blue text below the original comments from the reviewers.

Reviewer #2

I found the revised manuscript very clear and the response to every points raised by reviewers convincing. In particular, I think the broader description of the conceptual framework, with the inclusion of hypotheses H1 to H3, and their predictions P, as well as the concepts of clearance effort and clearance rate to account for significantly improved the paper. Although I found the figure 2 complex and a little overwhelming, I think the end of the introduction pages 7-8 explains it well.

This new figure 2 and the introduction of new clearance concepts serves the interpretation of the results and their discussion.

1. We are very happy to hear that our response to reviewers was satisfactory and that Figure 2 has helped understanding of the broader conceptual framework that our data fits into. Thank you for your positive feedback and suggestions. We very much appreciate them.

However, I think more details could be given about why data support H3 and not H2. Sometimes I was left unsure how the data could actually support H3, and I would have appreciated a couple of sentences around Lines 312-315 to help me navigate the results and tell me why it does not support H2.

2. Thank you for this suggestion. Just prior to "This finding supports our third hypothesis (H3)" formerly at line 315, we have added the following text: "This finding is inconsistent with prediction P2 of a positive relationship (Fig. 2j), i.e., that clearance rate increases with increasing exploitation, and it is instead consistent with prediction P3, i.e., that increased exploitation results in a lower clearance rate (Fig. 2n). Accordingly, this finding supports our third hypothesis (H3)." The new sentence can be found at lines 230-233.

We have also added the labels P1, P2 and P3 to figure 2 to hopefully aid with interpretation of this figure. And we have explicitly mentioned P3 and Fig 2k and n in the figure legend of figure 7, to hopefully also aid with interpretation of the results.

Minor comment:

The decision to represent a linear relationship between clearance rate and clearance effort in figure 1a, but an exponential relationship between immunity costs and clearance effort in 1c is not explicitly justified in the text. If I understand and mostly agree with this, I would have expected a line and/or a reference to back this up.

3. We have modified Figure 2 so that the relationships between immunity costs and clearance effort (1c) and host survival and clearance effort (1d) are now both linear. We think that this will make it easier to understand the figure. None of the predictions change as a result of altering the shapes of the relationships between the variables.

Reviewer #3

In this manuscript Acuña Hidalgo et al present work aiming to partition the variance in virulence between host clearance and pathogen persistence. They employ bacterial infections in the fruit fly *Drosophila melanogaster*, using different doses of 4 bacterial species and measured bacterial exploitation (from bacterial load), and per pathogen pathogenicity (PPP) from the relationship between virulence (measured from host mortality) and bacterial load, persistence and clearance of the bacteria.

They use this data to define and establish a unifying framework for infection outcomes, where pathogen clearance or pathogen persistence are the outcomes of a balance between the relative strengths of host clearance efforts, the per-pathogen pathogenicity, and the pathogen level of host exploitation. To me, this unifying framework is the major strength of this work and an immensely valuable contribution to the field of infection and immunity. Hopefully, the days of reading papers that start with “host can either resist or tolerate infection” are soon over, and this framework explaining that both are possible and necessary outcomes will take hold in the field.

The paper overall is very well-written, and I enjoyed the clever and creative experimental work, including the choice of bacterial pathogens that (intentionally) span a range of levels of virulence. I also appreciated the large amount of data, making the paper very complete, including data on the cost of resistance/clearance, which I do not believe has been demonstrated in this way previously.

I have the benefit of reviewing a version of the manuscript that has already been reviewed and revised, and so unsurprisingly, my criticism, if any is minor. I did read the authors' responses to previous queries and was pleased to see how well they were able to address them all.

My only minor criticism is the lack of any data on immune gene expression. To be clear, this should not be a requirement for publication of this paper, which makes a fantastic contribution to the field with the framework. Nor does it change the way the framework is interpreted. Simply it would have added useful mechanistic insight into the mechanisms of clearance in operation during the acute and persistent phases of infection. Though this may be something the authors would prefer to explore separately, in much greater detail, instead of being buried and lost in this paper, which is already so data-rich.

Overall, I loved the paper, congratulations.

4. Thank you for your positive feedback! We agree that it would be great to have some immune gene expression data from the acute and persistent infection phases. This is data that we plan to generate in the next couple of years as part of a recently started PhD project. As explained in our previous response to the reviewers, we feel that such immunity data is beyond the scope of the current manuscript.